# A Rapid Seismic Damage Assessment (RASDA) Tool for RC Buildings Based on an Artificial Intelligence Algorithm

Konstantinos Morfidis [1,*], Sotiria Stefanidou [2] and Olga Markogiannaki [2]

1 Earthquake Planning and Protection Organization (EPPO-ITSAK), Terma Dasylliou, 55535 Thessaloniki, Greece

2 Department of Civil Engineering, Aristotle University of Thessaloniki, Aristotle University Campus, 54124 Thessaloniki, Greece; sotiria.stefanidou@gmail.com (S.S.); markogiannaki.olga@gmail.com (O.M.)

* Correspondence: konmorf@gmail.com

**Abstract:** In the current manuscript, a novel software application for rapid damage assessment of RC buildings subjected to earthquake excitation is presented based on artificial neural networks. The software integrates the use of a novel deep learning methodology for rapid damage assessment into modern software development platforms, while the developed graphical user interface promotes the ease of use even from non-experts. The aim is to foster actions both in the pre- and post-earthquake phase. The structure of the source code permits the usage of the application either autonomously as a software tool for rapid visual inspections of buildings prior to or after a strong seismic event or as a component of building information modelling systems in the framework of digitizing building data and properties. The methodology implemented for the estimation of the RC buildings' damage states is based on the theory and algorithms of pattern recognition problems. The effectiveness of the developed software is successfully tested using an extended, numerically generated database of RC buildings subjected to recorded seismic events.

**Keywords:** seismic damage assessment; artificial neural networks; pattern recognition; software development; RC buildings

## 1. Introduction

High seismicity observed in many regions globally has always been a critical concern for the safety and prosperity of their communities. To avoid losses caused by the occurrence of strong earthquake events, researchers have been stimulated to develop solutions, such as seismic damage assessment methods, that provide increased and justified knowledge on the structural safety of the built environment exposed to seismic hazard. The application of seismic damage assessment methodologies mainly results in the evaluation of the expected damage level for a range of seismic intensities. This outcome could be either used in the pre- or post-earthquake assessment phase. In the pre-earthquake assessment phase, such methods are used to verify the design effectiveness and structures contributing to retrofit prioritization and planning optimum retrofit schemes. In the post-earthquake assessment phase, such results could be used to rapidly estimate the actual damage level of buildings after earthquake events in aid to first responders and civil protection authorities and to evaluate the existing resistance level against the expected seismic intensity.

Seismic damage assessment methods can be classified according to the anticipated complexity and the required computational effort. Several methodologies are available for the seismic assessment of structures, ranging from analytical to empirical and rapid visual screening (RVS) methods. Analytical methodologies require the development of appropriate mechanical models [1] and advanced analysis techniques for the estimation of the buildings' vulnerability [2], while empirical ones are based on damage statistics from past earthquakes [3] as well as on expert judgement [4]. The most common problem when applying a purely empirical approach is the unavailability of (sufficient and reliable)

statistical data for several intensities [5], leading to a relative abundance of statistical data in the intensity range from 6 to 8. However, it should be highlighted that purely analytical approaches should be avoided, since they might seriously diverge from reality, typically (but not consistently) overestimating the cost of damage [5]. Finally, the ATC-13 fragility curves based on expert judgement were found to grossly overpredict structural damage, at least for some classes of structure for which damage statistics were compiled [3]. Moreover, many methodologies are available worldwide for either pre- or post-earthquake rapid visual inspection of structures, while the most popular and widely used is the FEMA method for rapid visual screening (RVS) of buildings [6,7]. The NZSEE (New Zealand National Society for Earthquake Engineering) [8], JBDPA [9] and the GNTD [10] are also widely applied methodologies for rapid visual screening. The RVS procedure prescribed by FEMA uses a methodology based on the building survey inspection, and it requires the filling of a data collection form based on visual observation of the building from the exterior, and if possible, the interior.

As described above, several methods for rapid visual inspection (RVI) (or rapid visual screening (RVS)) are proposed [6–8], based on survey inspection and collection of data, thus not involving analyses and material strength justification. RVI methods are mainly based on visual observation and identification; therefore, they can be widely used for large-scale assessment (i.e., for entire cities or municipalities). However, in this manner, damage evaluation and assessment results strongly depend on expert judgment, thus affecting their effectiveness and reliability. The RVI methods are applicable for both pre-and post-earthquake assessment of structures contributing to the prioritization of interventions and development of emergency plans. It should be outlined that despite the inherent simplifications and assumptions, the RVI methods are valuable since they can be applied in a short time after an earthquake event, providing a first-stage evaluation of the damage extent and highlighting the need for detailed assessment and retrofit measures. The reliability of the assessment results is increased when RVI methods are used in conjunction with results of non-destructive tests and simplified calculations, while the highest level of reliability is achieved when the assessment is based on analysis of the detailed structural model highlighting the retrofit needs, also providing an optimum retrofit scheme and prioritization of interventions.

In line with the above, it is challenging to develop a framework, and a reliable tool for RVS of structures that could overcome the inherent disadvantages (mainly related to expert judgment) will be based on easily (in situ) collected data and require low computational effort. To this end, the current paper proposes a novel framework and a software application that exploits modern techniques and machine learning (ML) algorithms and is capable of extracting, in real-time, an estimation of the seismic damage assessment of individuals or groups of RC buildings either in the pre- or post-seismic phase. More specifically, the developed software tool is based on the expression of the seismic damage assessment as a pattern recognition (PR) problem [11,12]. This is a novel approach that differentiates the proposed software tool from other similar applications. By formulating the problem as a PR problem, the seismic damage assessment is achieved in a more supervisory manner and with less sensitivity to possible inaccurate data estimates, as the proposed methodology extracts the classification of buildings into damage categories directly and not through the quantitative estimation of a damage index.

The first step of RVI methods, either in the pre- or post-seismic phase, is the classification of structures into classes accounting for their seismic damage extent. This approach is compatible with the scope of the PR problems, which are dedicated to the classification of objects of the same type into pre-defined classes. The basic idea for the framework and software developed is to relate the macroscopic structural parameters (collected in the framework of RVI) and the seismic input parameters (based on selected accelerograms) to the expected seismic damage extent that is qualitatively and quantitively defined through seismic damage classes (SDC). Therefore, a preliminary rating of a structure is possible, which can be a valuable tool for rapid assessment and retrofit prioritization for both pre-

and post-earthquake assessment. Regarding the computational framework, the multilay-ered feedforward perceptron neural networks (MFPNN) were selected to solve the PR problem. Several research studies have highlighted the effectiveness of this type of artificial neural network (ANN) within the last three decades [13–19]. In addition, it should be noted that the use of ANNs as a computational tool in software development for the estimation of seismic damage assessment of structures has been proposed in recent years [20,21].

The software developed and presented herein includes a user-friendly and powerful graphic user interface (GUI), enabling the quick and easy insertion of the required structural and seismic input data. The features available include the graphical representation of useful structural and seismic input data parameters and the calculation report of relevant indexes defining the RC buildings' seismic damage state (SDC), thus enabling rapid damage assessment. The development of the software using modern development platforms [22,23] and the structure of the corresponding source code enables its wide implementation for pre- or post-earthquake seismic damage assessment of existing RC buildings. The software developed may form an independent module used locally (i.e., in desktops, laptops, or portable devices), or it can be included in integrated systems, such as "Building Information Modelling" (BIM) systems, which are the modern trend for the optimum design and management of structures, based on the digitization of building data and properties. Since BIM [24–27] is based on integrated systems covering all the stages of a building's life cycle, seismic damage prediction indices could be also incorporated. The inclusion of seismic damage assessment procedures in BIM systems has been proposed and investigated in several research studies and state-of-the-art papers [28–31]. The developed software includes a series of modern techniques that characterize and represent the next generation of the software application, dedicated to the rapid yet reliable and consistent seismic damage assessment in the pre- or post-earthquake phase of individuals or groups of RC buildings. The proposed framework using already-trained MFPNN [32,33] and the relevant software developed is successfully applied to selected case studies with either full or limited availability of the required structural property values.

The proposed methodology generally requires the existence of trained MFPNNs. For this training, it is necessary to use data dependent on the region in which the study is performed. This is not a limitation of the generality of the methodology as the formulation is generic and simply requires data that are applicable to each region (e.g., due to the validity of different design codes). The collection of training data is a relatively complex process but can be carried out either through a program involving numerical modeling and analysis or through data collection after a strong seismic excitation.

The results are presented and discussed herein, highlighting the effectiveness of the tool developed for the rapid seismic damage assessment of RC buildings using available RVI data.

## 2. Methodologies for Rapid Visual Inspection (RVI) of Buildings

Rapid visual screening (RVS) of buildings is a methodology used to quickly assess the potential seismic hazards of buildings. It involves a visual inspection of a building's exterior and interior to identify potential hazards and damage, such as structural damage, non-structural damage, and functional damage. Overall, it is a practical and cost-effective tool for assessing the seismic hazards of buildings and identifying retrofitting needs. It is widely used by building professionals, engineers, and government agencies to prioritize their seismic risk reduction efforts and ensure the safety of the built environment. RVS is often used in emergency situations to quickly assess the safety of a building and prioritize its repair or retrofitting needs. The Federal Emergency Management Agency (FEMA) devel-oped the RVS methodology in the 1990s, and it has since been widely adopted by building professionals, engineers, and government agencies [6,7]. FEMA's *Rapid Visual Screening of Buildings for Potential Seismic Hazards: A Handbook* provides guidance on how to conduct an RVS assessment, including the use of a building checklist, identification of potential haz-ards, and prioritization of retrofitting needs. RVS assessments can be performed by trained

professionals or building occupants with basic training. The methodology is cost-effective and requires minimal equipment, making it a practical option for rapid assessment of large numbers of buildings.

However, RVS assessments may not provide as much detailed information as other methods, and their accuracy may be limited by the expertise of the assessor and the quality of the data collected. RVS may be used for prioritization of building retrofitting needs, identifying the most hazardous buildings in a community, and prioritizing retrofitting needs. The assessment can help building's owners and managers make informed decisions on which buildings to retrofit first and allocate resources accordingly. Furthermore, RVS can be used for emergency response planning to identify buildings that may be at risk of collapse or severe damage in the event of an earthquake as well as for risk assessment of building stocks, identifying high-risk buildings, and intervention needs. Many methodologies are available worldwide for either pre- or post-earthquake rapid visual inspection of structures, while the most popular and widely used is the FEMA method for rapid visual screening (RVS) of buildings [6,7]. The New Zealand National Society for Earthquake Engineering [8], the Japan Building Disaster Prevention Association [9], and the GNTD [10] are also widely applied methodologies for rapid visual screening. The RVS procedure prescribed by FEMA uses a methodology based on the building survey inspection, and it requires the filling of a data collection form based on visual observation of the building from the exterior and, if possible, the interior. The two-page data collection form includes building identification information (i.e., usage, area, floor number, etc.), a photograph of the building, sketches, and documentation of pertinent data related to seismic performance.

Simple survey procedures for seismic risk assessment are proposed and applied to urban building stocks to provide damage statistics [34]. In most cases, the expected damage extent is classified as slight, moderate, major, and collapse according to the indicative qualitative definitions of Table 1, based on FEMA.

**Table 1.** Classification and description of seismic damage according to FEMA.

| Damage State | Qualitative Description |
|---|---|
| *Slight damage* | No permanent drift. Structure substantially retains original strength and stiffness. Minor cracking of facades, partitions, and ceilings as well as structural elements. All systems important to normal operation are functional. |
| *Moderate damage* | Wider cracks at non-structural elements, in-plane or out-of-plane. Some residual strength and stiffness are left in all stories. Gravity load-bearing elements function. No out-of-plane failure of walls or tipping of parapets. Some permanent drift. Damage to partitions. Cracking of facades, partitions, and ceilings as well as structural elements. Flexural and shear cracks at structural elements, concrete spalling. |
| *Major damage* | Local in-plane and out-of-plane failure of nonstructural walls, infills, etc. Wide flexural and shear cracks at structural elements, hoop fracture, buckling of longitudinal reinforcement, initiation of concrete core crushing. Little residual stiffness and strength, but loadbearing columns and walls function. Large permanent drifts. Some exits are blocked. Infills and unbraced parapets failed or at incipient failure. |
| *Collapse* | Loss of load-carrying capacity, locally or globally. |

## 3. The Proposed Method for the Rapid Seismic Damage Assessment of RC Buildings

In recent years, there has been a growing trend of using artificial intelligence techniques to create tools that can assist engineers, architects, and policymakers in making informed decisions about building safety. The aim is to develop tools that are accurate, scalable, and user-friendly, with the potential to enhance the effectiveness of current practices in these fields. The overall vision of using machine learning (ML) in rapid visual screening (RVS) for buildings is to leverage the power of data-driven models to improve the accuracy and

efficiency of the screening process. By integrating ML techniques into RVS, it is possible to automate and standardize the screening process, reducing the reliance on human judgment and improving the consistency of the results. ML algorithms can be trained on large datasets of building characteristics and historical seismic data to identify patterns and correlations that can be used to predict the seismic performance of buildings. This can help prioritize buildings for more detailed assessment and retrofit, leading to a more targeted and efficient use of resources.

The rapid seismic damage assessment method proposed is based on the formulation and solution of a pattern recognition (PR) problem. The PR is one of the problems that can be solved using machine learning (ML) algorithms [11,12]. For this reason, artificial neural networks (ANN) and, more specifically, multilayered feedforward perceptron neural networks (MFPNN) are implemented. The theory and the applications of the MFPNN are described in detail in several books (e.g., [35–37]).

The first step towards the development of a methodology for the damage assessment of RC buildings using PR and MFPNN is the definition of the relevant input and output parameters, considering the basic principles of the PR problems and the solution procedure using MFPNN. By definition, PR is the procedure for detecting and classifying objects of the same type into specific classes. A PR problem can be defined using three different approaches: the supervised learning approach, the unsupervised learning approach, or the reinforcement learning approach [37]. The software application developed herein applies the *supervised learning approach*. Therefore, the target of the applied algorithm is the consistent classification of objects into pre-defined classes. For this reason, the creation of an appropriate dataset and the training of a properly configured MFPNN using this dataset are required. The general configuration of an MFPNN, which can be used to solve a PR problem with *n* pre-defined classes implementing the supervised learning approach, is depicted in Figure 1. The population of the input parameters, i.e., the parameters that describe the objects (elements of the input vectors **x**) is *m*. Thus, in this case, the target is the classification of objects described using the **x** vectors (having *m* parameters) into *n* predefined classes.

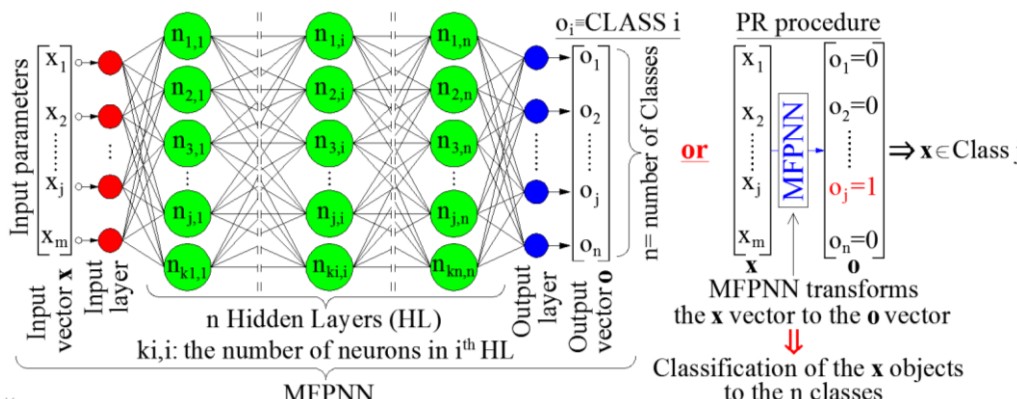

**Figure 1.** General description of the configuration of an MFPNN used for the solution of PR problems.

In order to use PR techniques and ML algorithms for rapid seismic damage assessment, the following should be applied:

(a) The type and number of the characteristics of the objects (i.e., the input **x** vectors) should be selected and classified into pre-defined classes. Furthermore, the corresponding classes' number and type should also be defined (i.e., the output **o** vectors). An important issue, which is also briefly described in Figure 1, is the mapping of output vectors ($o_j$ elements) to the classes of the problem. More specifically, the configuration of **o** vectors for each one of the pre-defined classes must be defined. By definition, when the MFPNN extracts an output vector **o** with $o_j = 1$ and all the other elements of **o** are equal to 0, then the corresponding object (**x** vector) is classified to

the class *j*. For the seismic damage assessment, the input vectors **x** should contain parameters that are crucial for the seismic performance of the RC buildings (structural parameters), as well as parameters that describe the seismic excitation (seismic parameters) [38]. To this end, the input vector **x** consists of two sub-vectors, namely the sub-vector $\mathbf{x}_{struct}$ and the sub-vector $\mathbf{x}_{seism}$ (Equation (1)):

$$\mathbf{x} = \left[ \mathbf{x}_{seiss} \mid \mathbf{x}_{struct} \right]^{\mathrm{T}} \tag{1}$$

(b)  The structural and seismic parameters should be selected. Since the target of the current paper is the development of the software application based on the PR approach for the rapid seismic damage assessment of RC buildings, no further investigation regarding the optimum selection of the structural and the seismic parameters was performed. Moreover, as explained in the next section, the structure of the software code allows for modifications in the case of MFPNNs with different input parameters (structural and seismic). At this stage of development, already trained and successfully tested MFPNNs were used, while the description of the parametric investigation for the optimum configuration of these MFPNNs is given in [32]. A brief description of the selected input and output parameters of the introduced MFPNNs as well as their configuration parameters are presented herein. Regarding the selection of the *structural parameters*, four parameters were selected to consider parameters that are critical for the seismic performance, and these are also considered in the framework of taxonomy and classification systems proposed for fragility assessment. The parameters selected are the total height of the building ($H_{tot}$), the ratio of the base shear that is received by RC walls along two perpendicular directions between them: directions 1 and 2 (ratio $n_{v1}$ and ratio $n_{v2}$), and the structural eccentricity $e_0$ (i.e., the distance between the mass center and the stiffness center of stories). However, it must be noted that the estimation of the parameter's values could be difficult in the case of the RVI. For this reason, as it will be presented in Section 4.4, the developed software allows the input of user-defined parameters, provided that the values are known from previous studies or measurements. In case the values are unknown, a parametric investigation is automatically performed (considering a realistic range of values for specific input parameters) to account for the effect of their variation on the classification of the examined building. This feature renders the software applicable to buildings with both reliable known and unknown (or non-reliable known) structural properties; however, the parametric investigation introduces an inherent uncertainty that is generally acceptable in the framework of the rapid seismic damage assessment methods. The selected *seismic parameters* are well-documented parameters for the description of the seismic excitations [39,40], widely and effectively used in several research studies [41]. The seismic parameters used herein are summarized in Table 2.

**Table 2.** The seismic parameters of the input vectors for the trained MFPNNs of the developed software.

| Seismic Parameter | | | |
|---|---|---|---|
| 1 | Peak Ground Acceleration—**PGA** | 8 | Housner Intensity—**HI** |
| 2 | Peak Ground Velocity—**PGV** | 9 | Effective Peak Acceleration—**EPA** |
| 3 | Peak Ground Displacement—**PGD** | 10 | $V_{max}/A_{max}$ (**PGV/PGA**) |
| 4 | Arias Intensity—$\mathbf{I_a}$ | 11 | Predominant Period—**PP** |
| 5 | Specific Energy Density—**SED** | 12 | Uniform Duration—**UD** |
| 6 | Cumulative Absolute Velocity—**CAV** | 13 | Bracketed Duration—**BD** |
| 7 | Acceleration Spectrum Intensity—**ASI** | 14 | Significant Duration—**SD** |

The final form of the input vectors **x** is given in Equation (2).

$$\mathbf{x} = \left[ \mathbf{x}_{\text{seism}} \mid \mathbf{x}_{\text{struct}} \right]^T$$

$$\mathbf{x}_{\text{seism}} = \left[ \text{PGA} \mid \text{PGV} \mid \text{PGD} \mid I_a \mid \text{SED} \mid \text{CAV} \mid \text{ASI} \mid \text{HI} \mid \text{EPA} \mid \text{PGV/PGA} \mid \text{PP} \mid \text{UD} \mid \text{BD} \mid \text{SD} \right]^T \quad (2)$$

$$\mathbf{x}_{\text{struct}} = \left[ H_{\text{tot}} \mid e_0 \mid n_{v1} \mid n_{v2} \right]^T$$

(c)　The seismic damage classes (SDC) should be qualitatively and quantitively defined considering appropriate engineering demand parameters (EDPs) and relevant threshold values. The EDPs (which in the present case are also defined as seismic damage indices (SDI)) could be either global or local [42]. Threshold values should be defined to highlight damage initiation for the limit state considered. The maximum interstory drift ratio (MIDR), which is an SDI that refers to buildings' global performance, is selected herein as EDP. Several (5 to 3) damage states (DS)—which are mapped to SDCs—are proposed in the literature for RC buildings, as well as the relevant threshold values, as presented in Table 3 [43]. The MFPNNs used within the software code were trained using the maximum interstory drift ratio (MIDR). The SDCs considered in the framework of the proposed approach are three, having the threshold values shown in Table 3.

**Table 3.** Definition of the SDC (DS) using the MIDR as EDP.

| MIDR [%] | <0.25 | 0.25–0.5 | 0.5–1.0 | 1.0–1.5 | >1.5 |
|---|---|---|---|---|---|
| SDC (5 classes) | Null | Slight | Moderate | Severe | Collapse |
| SDC (3 classes) | Slight ("S") | | Moderate ("M") | Severe/Collapse ("S-C") | |
| Description | No (or repairable) structural damages | | Significant but repairable structural damages | Severe or non-repairable structural damages | |

As already stated, the MFPNNs used in the current version of the developed software were trained using three SDCs. This approach is more compatible with RVI methods (as well as with the rapid damage assessment methods carried out after a strong earthquake), where the characterization of the seismic damage of buildings is not detailed. The MFPNNs were trained using three different training datasets created for three RC building classes, classified according to masonry infills' existence. In particular, the classes considered are (a) buildings without masonry infills or with light masonry infills (bare buildings, BB), (b) buildings with masonry infills at all stories (masonry buildings MB), and (c) buildings with masonry infills at all stories except the ground story (buildings with pilotis, PB). More details about the training procedure of these networks are given in [33]. Based on the above, the current version of the software has built-in three trained MFPNNs (see also Section 4.2.4). The optimum configuration of the used MFPNNs is presented in Figure 2.

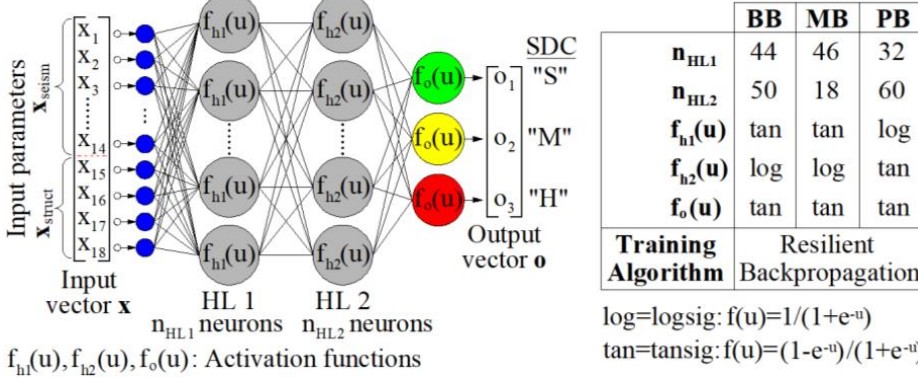

**Figure 2.** Configuration parameters of the optimum MFPNNs used in the current version of the software.

## 4. Description of the Developed Software

### 4.1. General Description

A detailed description of the software application is presented in the current section. The software was developed on the MATLAB platform [22,23] and is currently provided as an executable file, while the development of a relevant online tool in Python is currently underway. The main advantages of the MATLAB and the Python software development platforms are their tools for the programming of effective and functional GUIs, as well as the inclusion of machine learning (ML) algorithm libraries required for the software proposed. An overview of the software application is presented in Figure 3. A comprehensive and detailed GUI is developed, including all the necessary features for a prompt and efficient operation (described in Section 4.2). The software was developed on the basis of event-driven programming; therefore, the user controls all the procedures through appropriate functions that are activated using the GUI's control components. More details about these control components are presented in Sections 4.2 and 4.3. A detailed description of the procedures is provided in Figure 3.

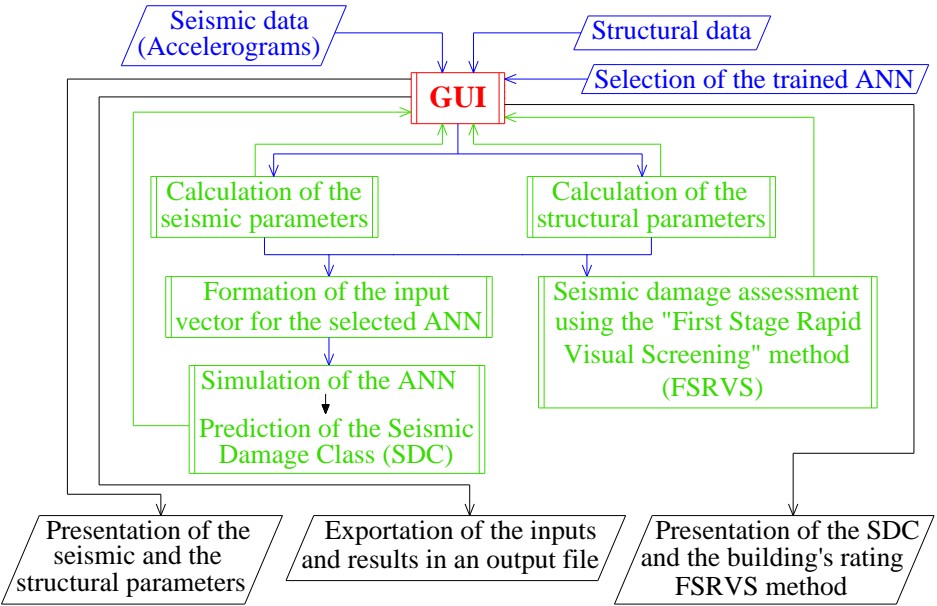

**Figure 3.** Overview of the developed software application.

The developed software provides (a) prediction of the expected DS (SDC) of RC buildings subjected to seismic ground excitations (considering two horizontal components of the corresponding seismic records) using properly trained ANNs that can be alternatively added by the user and (b) estimation the buildings' seismic damage rating using the first stage rapid visual screening (FSRVS) procedure, which is practically an RVI methodology proposed by Earthquake Planning and Protection Organization of Greece (E.P.P.O.) for pre-earthquake assessment [44]. Furthermore, the software provides (as intermediate results) the parameters that describe the selected seismic excitation (spectra and seismic parameters) and properly selected structural parameters that are critical for the seismic damage assessment of a studied RC building.

The software is outlined in the sections below. In particular:

- Description of the GUI and its components used for the insertion of the required data and the presentation of the results.
- Description of the source code, i.e., the functions used and the interaction between them, along with the required data processing and the corresponding flowcharts.

*4.2. Description of the GUI's Components*

The proposed application is controlled through a basic/main window organized in four panel containers (Figure 4). These panels correspond to four different procedure categories:

- Panel 1 entitled "INPUT DATA" contains the GUI tools/components for the structural and seismic data input.
- Panel 2 entitled "SEISMIC PARAMETERS and SPECTRA" includes the presentation of the spectra and the seismic parameters of the selected input seismic excitation.
- Panel 3 entitled "STRUCTURAL PARAMETERS" includes the illustration of the elastic spectrum of the RC building studied and the structural parameters that are crucial for the seismic damage assessment in the framework of RVI.
- Panel 4 entitled "RESULTS" presents the final results of the software, i.e., DS (SDC) prediction (extracted by the selected ANN) for the RC building subjected to the input seismic excitation, as well as its seismic damage assessment and prioritization based on the rating system proposed within the FSRVS procedure.

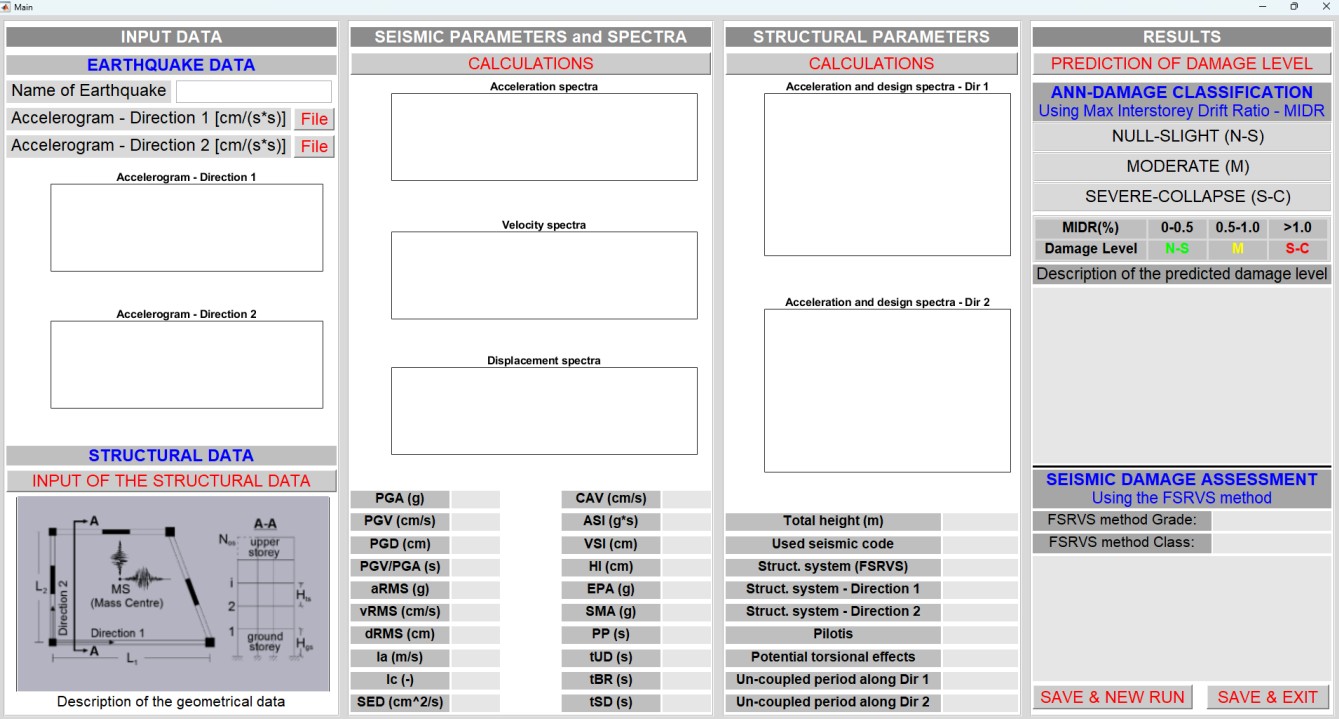

**Figure 4.** Main window of the proposed application (before the insertion of the input data).

The two push buttons entitled "FILE" activate a window with a file selector for the selection of the accelerograms of the earthquake excitation (in text file format) in directions 1 and 2 (Figure 6).

### 4.2.1. Panel "INPUT DATA"

The panel "INPUT DATA" contains the GUI tools for the seismic and structural data insertion. Figure 5 presents the procedures performed by the GUI tools of the panel.

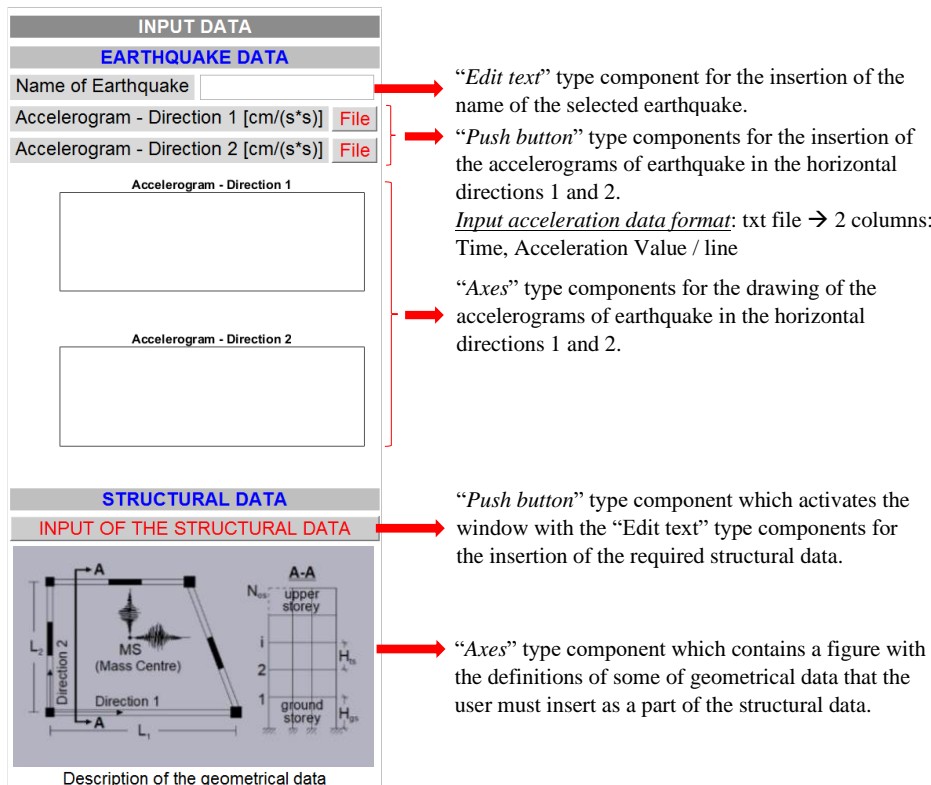

**Figure 5.** Brief description of the GUI control tools/components of the panel "INPUT DATA".

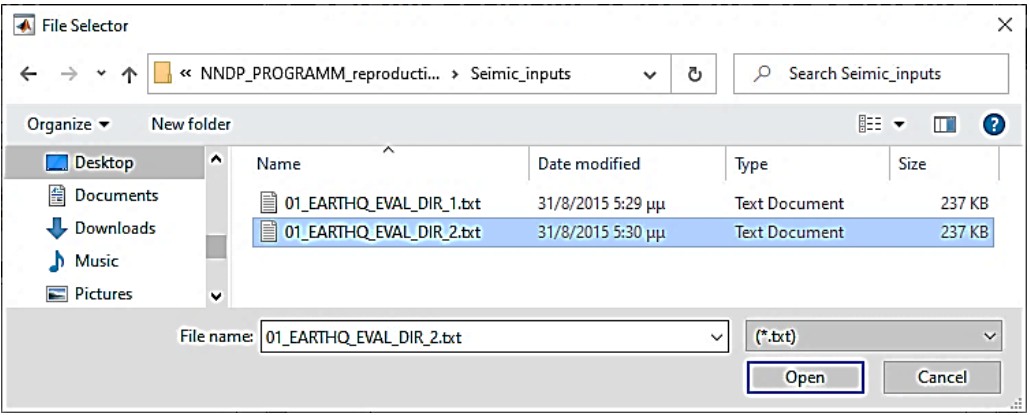

**Figure 6.** The file selector activated when the two push buttons "FILE" are pushed by the user.

The push button entitled "INPUT OF THE STRUCTURAL DATA" activates a window that contains the appropriate GUI control tools/components (i.e., "static text", "edit text", and "popup menu") for the insertion of the required structural data (Figure 7a). It should be outlined that the inserted structural data are used for the formation of the input vector for the trained ANNs and the DS prediction as well as for the seismic damage assessment and prioritization of the RC building studied, in line with the FSRVS rating system proposed by E.P.P.O. [44].

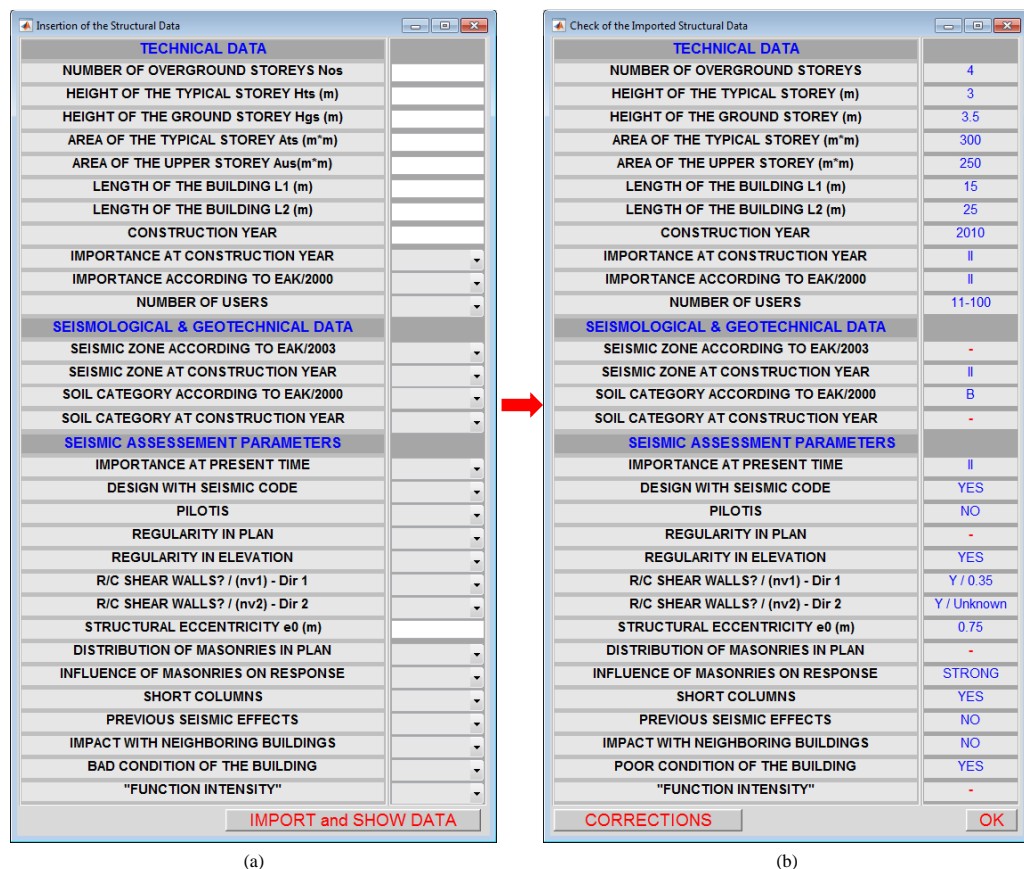

**Figure 7.** The windows used for the insertion (**a**) and checking (**b**) of the structural data.

Using the push button entitled "IMPORT and SHOW DATA", the window of Figure 7b is activated. This window contains the structural data inserted by the user. Thus, an additional check for erroneous or omitted data can be performed. Towards this check, the values of inserted data are presented using blue fonts and the omitted data using red hyphens. If the user confirms that all the inserted data are correct, the push button "OK" finishes the procedure. Then the two windows of Figure 7 are closed, and the inserted data are stored in a matrix for further calculations. In case errors are detected, the push button "CORRECTIONS" is available for the relevant corrections. Then, the user should go back to the window of Figure 7a and proceed with the corrections.

4.2.2. Panel "SEISMIC PARAMETERS and SPECTRA"

This panel is the first among the two panels used to present the intermediate results. More specifically, the panel "SEISMIC PARAMETERS and SPECTRA" is related to the presentation of the seismic and spectral parameters of the selected seismic excitations. As shown in Figure 8, the panel contains the push button entitled "CALCULATIONS", which activates the connected external function (see Section 4.3) to calculate the acceleration, velocity, and displacement spectra for the two horizontal components of the seismic excitation (directions 1 and 2) and the corresponding (20) selected seismic parameters [39,40]. Diagrams with the calculated spectra are provided within the panel, along with the seismic parameters presented in "static text" GUI components (Figure 8).

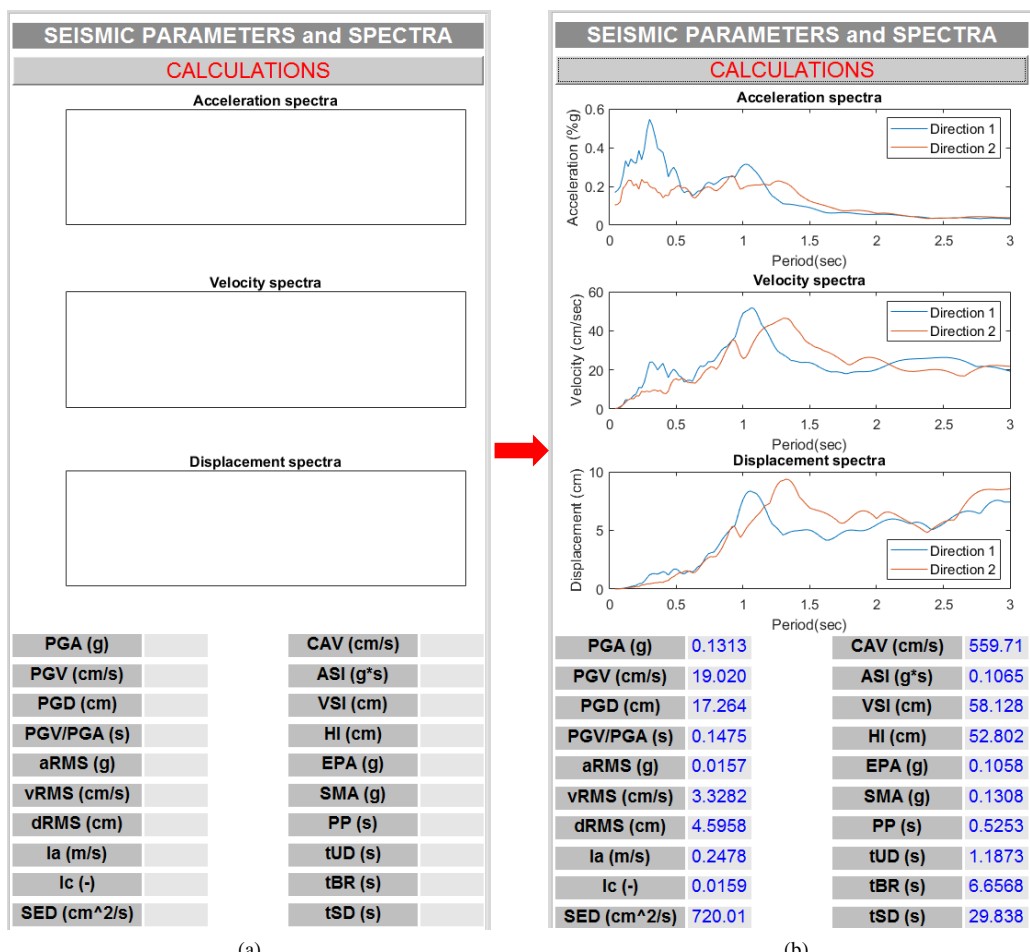

**Figure 8.** The window used for the presentation of the seismic parameters (**a**) before and (**b**) after running.

It should be noted that the values of the seismic parameters presented in the panel are the geometric means of their values extracted from the input accelerograms along the two horizontal directions. However, the seismic parameters corresponding to the individual accelerograms are also printed in the output file (in .txt or .html format) along with the input data and the other results (Figure 3).

### 4.2.3. Panel "STRUCTURAL PARAMETERS"

This panel (Figure 9) is the second among the two panels used to present the intermediate results, i.e., the selected structural parameters. In order to run the corresponding external function (see Section 4.3) and depict these parameters, the push button entitled "CALCULATIONS" should be used.

The most indicative representation of this panel is the depiction of (a) the elastic acceleration spectrum of the selected excitation, (b) the elastic design spectrum of the studied RC building, and (c) the predicted range of the building's uncoupled fundamental elastic eigenperiods, in the same figure (Figure 9b). The parameters above are estimated for the two horizontal directions 1 and 2 and are presented in separate figures. Finally, it should be outlined that the equations used for the estimation of the values of the uncoupled fundamental elastic eigenperiods of the studied buildings are well-documented formulas for RC buildings with or without RC shear walls [45–47].

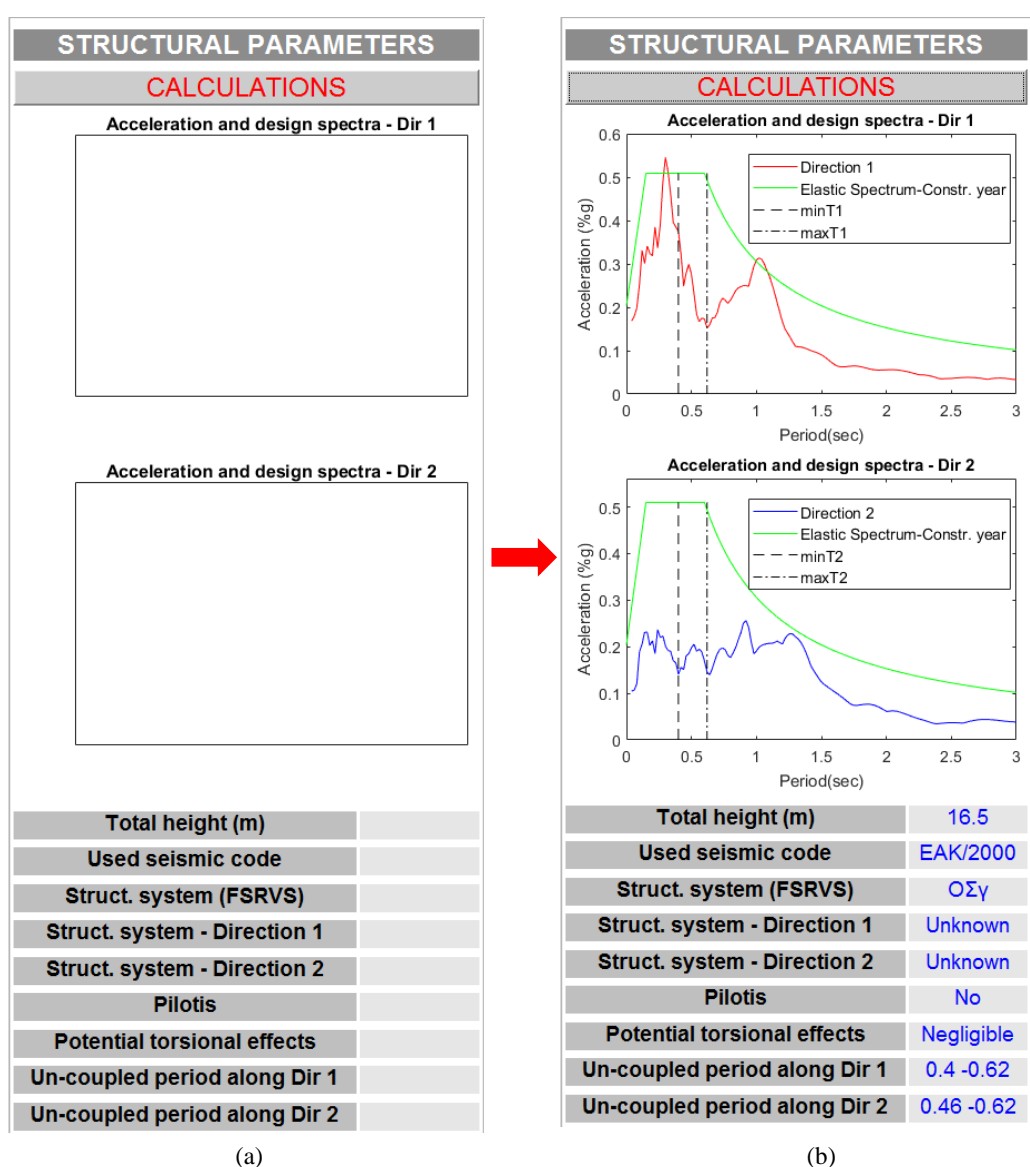

**Figure 9.** The window used for the presentation of the structural parameters (**a**) before and (**b**) after running.

### 4.2.4. Panel "RESULTS"

The panel presented in Figure 10 is the last panel depicting the final results of the application and extracting the output file.

When the push button entitled "PREDICTION OF DAMAGE LEVEL" is used, the window of Figure 11 is activated. This window enables the selection of the trained MF-PNN that will be used to estimate the DS (SDC) of the studied RC building for the input earthquake ground motion selected.

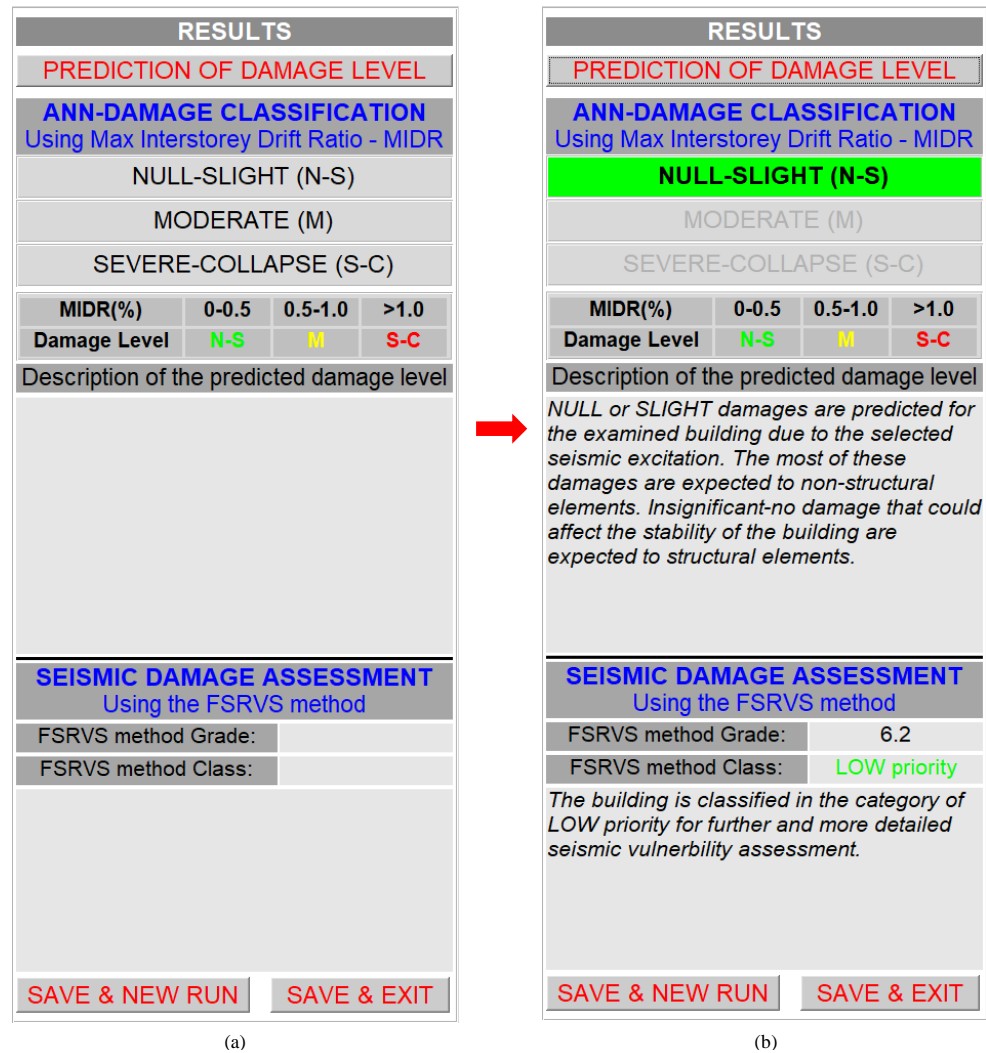

**Figure 10.** The window used for the presentation of the final results (**a**) before and (**b**) after running.

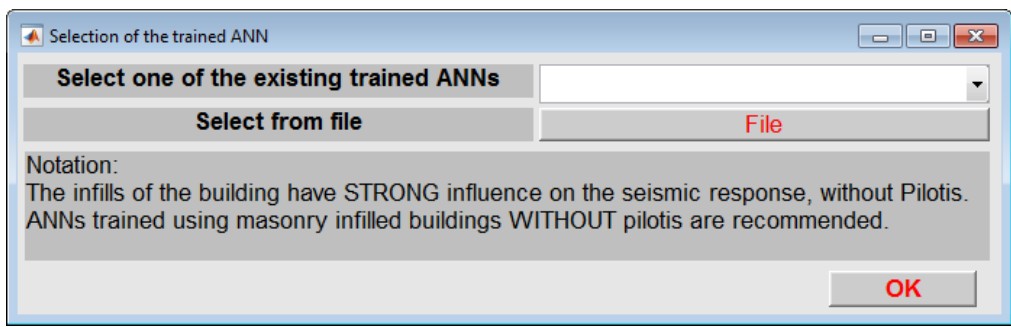

**Figure 11.** The file selector activated for the selection of the trained MFPNN.

Two general options are available:

(a)    Selection of one of the three, embedded in the software application, trained MFPNNs using a pulldown menu. Each one of these MFPNNs was trained considering three general classes of RC buildings regarding the masonry infills. Buildings without masonry infills or with light masonry infills, buildings with masonry infills at all stories, and buildings with masonry infills at all stories except for the ground story (buildings with pilotis) were used. These embedded MFPNNs were trained and successfully applied as presented in [32,33]. It should also be noted that these MFPNNs were

trained with a database generated from non-linear dynamic analyses of RC buildings designed according to Eurocodes.

(b) Selection and importation of an MFPNN trained by the user. For this purpose, the push button "File" is available, activating a window with a file selector. The selected file should be compatible with the MATLAB files/objects containing trained ANNs (".mat" type files). In addition, the imported MFPNNs should be compatible with the MFPNNs used in the current version of the software application (Section 3) as regards the number and the types of the input and output parameters. After the introduction of a user-selected trained MFPNN, the application functions as in the case where one of the embedded MFPNNs is selected.

In Figure 11, detailed instructions for selecting the most suitable ANN for predicting the DS of the studied RC buildings are provided. These instructions are in line with the structural data inserted by the user (Figure 7). For example, suppose that structural parameters inserted by the user for the RC building studied are related to the existence of masonry infills at all stories except the ground story (i.e., RC building with pilotis), a notation will then appear in the window, suggesting the implementation of a compatible ANN (i.e., the implementation of an ANN trained using the dataset created for RC buildings with pilotis).

After selecting the trained ANN, the software activates the function that simulates this network and extracts the prediction regarding the SDC (i.e., the DS) of the studied RC building (see Section 4.3). This prediction is presented in the properly designed field of the panel (Figure 10). In addition, a short description of the extracted DS is presented in the corresponding static text area (Figure 10). The software also activates the function that performs all the required calculations for the rating of the studied RC building according to the FSRVS procedure. This rating as well as the classification of the studied building in categories with a view to prioritizing interventions (i.e., low, medium, or high) are also presented in the panel (Figure 10).

Finally, the panel contains two push buttons, entitled "SAVE and NEW RUN" and "SAVE and EXIT". Each one activates a different function (see Section 4.3). The first one activates a function that creates an output file (in .txt or .html format), which includes all input data and the results, cleaning the memory for a new run. In contrast, the second one activates a function that also creates an output file with the data provided and the results produced and closes the application. The output could be incorporated even in building information modelling systems.

### 4.3. Description of the Structure of the Source Code

The source code of the application was developed on the basis of a central/main function that controls the other components/functions of the program (Figure 3). The basic operation of the main function is the programming of the main window and all the other windows of the GUI. In addition, the main function controls all functions of the program, which perform the required calculations, and manages the transferring of the data between them. In other words, the main function controls the insertion of the user data, calls the functions that (using the input data) calculate the intermediate and the final results, and manages their representation. The code of the main function is organized through different code "blocks", which perform specific operations. These "blocks" are summarized in Figure 12.

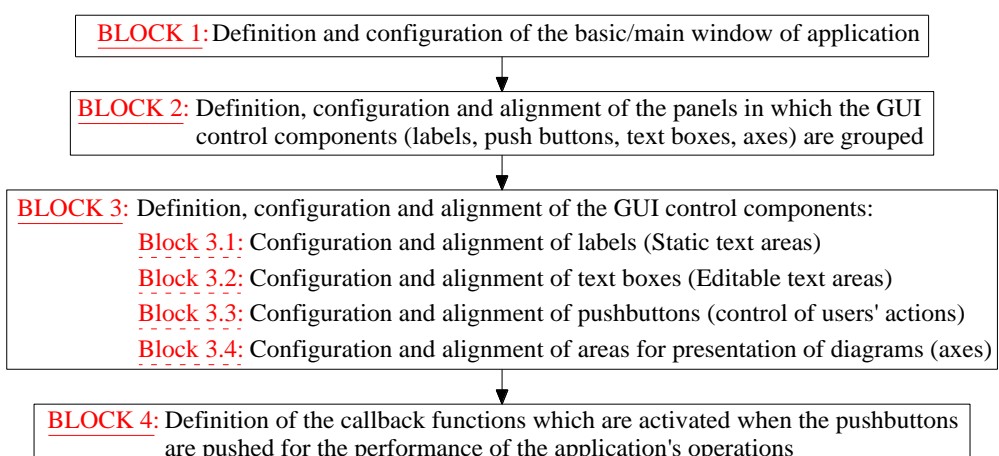

**Figure 12.** General description of the structure of the software application's main function.

The functions are separated into two main categories:

(a) The functions embedded in the main function (internal functions). These functions are used to develop the user interface push buttons, i.e., the operations performed when the user calls them (callback functions). A brief description of these functions is presented in Table 4.

(b) The external functions used from the callback functions. These functions perform specific procedures, which are mainly computational. The external functions contain the program code, leading to the intermediate and final application results (Figure 3). A brief description of these functions is presented in Table 5.

**Table 4.** Description of the internal functions of application (built-in in the file of the main function).

| Function/[GUI Component—Panel] | Procedure |
|---|---|
| Acc1_Callback/[Pushbutton "FILE"—Panel 1], (Figure 5) {Acc2_Callback/[Pushbutton "FILE"—Panel 1]}, (Figure 5) | Insertion (through the activation of a window with file selector), saving, and plotting of the accelerogram of the selected excitation in the direction 1 {2}. |
| StrInpData_Callback/["INPUT OF THE STRUCTURAL DATA"—Panel 1], (Figure 5) | Creates and activates a GUI window for the insertion of the structural data (this window activates a second window for the checking of the imported data using the function ChStrDat Callback activated by the push button "IMPORT and SHOW DATA"). |
| SePar_Callback/["CALCULATIONS"—Panel 2], (Figure 8) | Calculates (using the external function "SEISMIC_PARS") and presents of the excitation's acceleration, velocity and displacement spectra and the seismic parameters. |
| StrPar_Callback/["CALCULATIONS"—Panel 3], (Figure 9) | • Calculates and plots the excitation's acceleration spectrum and the design elastic acceleration spectrum (external function "GR_CODE_EL_SPECTR"). <br> • Calculates the structural parameters used for the rapid estimation of the building's seismic damage assessment. |
| Results_Callback/["PREDICTION OF DAMAGE LEVEL"—Panel 4], (Figure 10) | Classifies the building to one of the three pre-defined DS (SDC) and its rating according to the FSRVS procedure proposed by E.P.P.O. [by calling the external functions: "TRAINED_ANN SELECTOR", "ANN_CALCS" and "FSRVS_CALCS"]. |
| SavEx_Callback/["SAVE & EXIT"—Panel 4], (Figure 10) | • Calls the external function "FINAL_OUTPUT" that creates the output file which includes all input data and results. <br> • Closes the application. |

**Table 4.** *Cont.*

| Function/[GUI Component—Panel] | Procedure |
| --- | --- |
| SavNRun_Callback/["SAVE & NEW RUN"—Panel 4], (Figure 10) | • Calls the external function "FINAL_OUTPUT" that creates the output file which includes all input data and results.<br>• Clears the memory (external function "CLEAR_MEM") and restarts the application. |

**Table 5.** Description of the external functions of the software.

| Function/Called from Callback Function | Procedure |
| --- | --- |
| "SEISMIC_PARS"/SePar_Callback | Calculates the seismic parameters of the selected excitation in directions 1 and 2. |
| "GR_CODE_EL_SPECTR"/StrPar_Callback | Calculates the design elastic acceleration spectrum (according to the valid seismic codes at the year of the building's design). |
| "FORM_ANN _INPUT"/Results_Callback | Configuration of the vectors used as input of the selected trained ANNs. |
| "TRAINED_ANN_SELECTOR"/Results_Callback | Creates and activates a window with GUI tools for the selection and insertion of a trained ANN (Figure 11). |
| "ANN_CALCS"/Results_Callback | Simulates the imported ANN for the rapid prediction of the DS of building. |
| "FSRVS_CALCS"/Results_Callback | Required calculations for the rating of building and its classification into the priority classes, defined in the framework of FSRVS method. |
| "FINAL_OUTPUT"/SavEx_Callback and SavNRun_Callback | (a) Creates and activates a window for the saving of the final output file.<br>(b) Creates the output file that includes all input data and (intermediate, final) results. |
| "CLEAR_MEM"/SavEx_Callback and SavNRun_Callback | Clears the memory from the parameters that are defined during the running of the application. |

Figure 13 briefly describes the interconnection between the internal (Table 4) and the external (Table 5) functions. The transmission of data between the functions starting from the data input to the final presentation of the results on the screen and the creation of the final output file are also presented.

### 4.4. Methodology for the Estimation of Unknown Input Structural Parameters

As mentioned in Section 4.3, the input vectors of the MFPNN used are formed utilizing the external function "FORM_ANN_INPUT" (Table 5). Moreover, as stated in Section 3 (Equation (2)), the input vectors of the MFPNNs embedded in the current version of the software application include 14 seismic (Table 2) and four structural parameters (Equation (2)).

The seismic parameters are estimated by the processing of the input records (accelerograms) that could be obtained from freely available databases (e.g., PEER strong motion database [48]) or in real-time (immediately after an earthquake event) by the stakeholders of accelerometric stations using well-documented formulas [39,40]. These formulas are inserted in the external function "SEISMIC_PARS" (Table 5).

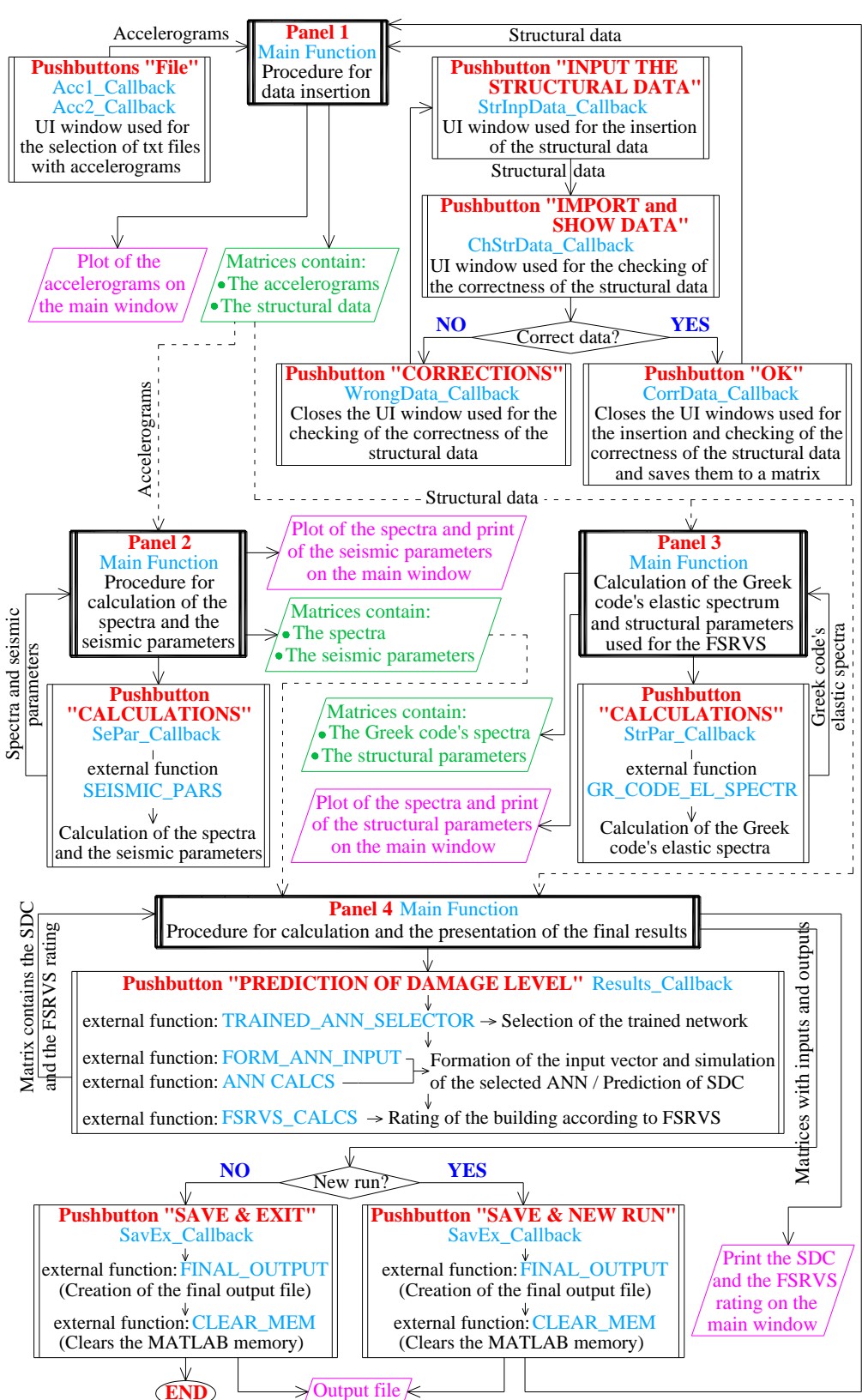

**Figure 13.** Brief flow chart of the source code of the developed software application.

On the contrary, the structural parameters required as input are, in general, difficult to estimate instantly since they are not limited to geometrical parameters (such as the total height of buildings H$_{tot}$, Figure 14) that could be easily measured or defined during RVI.

Input parameters such as the base shear ratio that is received by RC walls (if they exist) along the two principal directions (ratios $n_{v1}$ and $n_{v2}$) and the structural eccentricity $e_0$ require time-consuming calculations (not compatible with the RVI methods) in order to be defined.

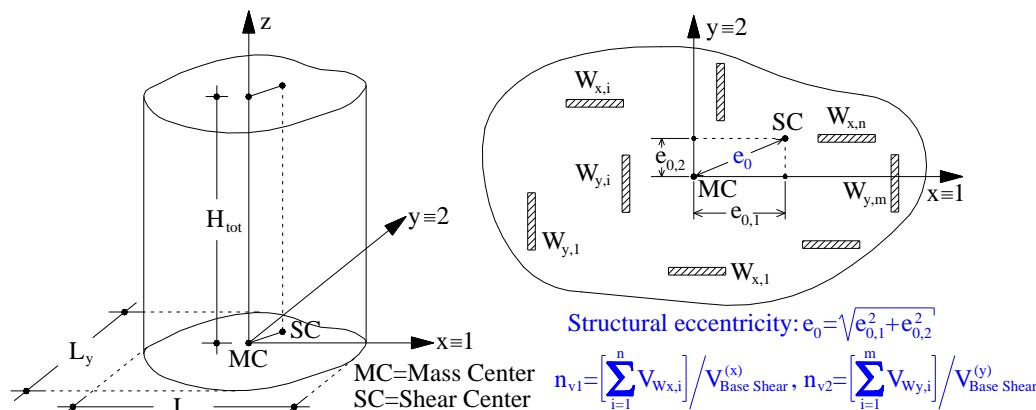

**Figure 14.** Description of the structural parameters imported to the input vectors of the trained MFPNNs.

As already mentioned in Section 3, the software application proposed herein can be implemented for the seismic damage assessment of RC buildings independently of reliable data availability. Therefore, the structural parameters of the MFPNNs' input vectors (Equation (2)) could be reliably known (measured in situ, obtained by technical reports and drawings, or calculated) or completely unknown. In the first case, the user can directly insert the values of the four required structural parameters. In the second case, the user defines the missing parameters as unknown. Reasonable assumptions (based on the available data (Figure 7)) are necessary for the use of the software developed, since it is based on MFPNNs that require the input of the aforementioned structural parameters (Equation (2)). The parameters that are difficult to estimate, using the available structural input data, are the ratios $n_{v1}$ and $n_{v2}$ and the value of the structural eccentricity $e_0$.

Based on the above, parametric analyses are proposed in order to apply the software developed consistently and reliably, using the external function "FORM_ANN _INPUT" (Table 5). The first approximation of $n_{v1}$, $n_{v2}$, and $e_0$ values, is a reasonable, user-defined value range (i.e., minimum and maximum values). This value range is estimated accounting for the structural data inserted by the users to the corresponding GUI window (Figure 7).

Regarding the structural eccentricity $e_0$, the minimum ($mine_0$) and the maximum ($maxe_0$) are defined based on the following two structural data inserted by the user (Figure 7): (a) Regularity (or not) of the building in plan, and (b) Regular (or not) distribution of masonry infills in plans. According to the input of the aforementioned structural parameters, three levels of structural eccentricity are defined, namely low, medium, and high eccentricity. These levels are related to specific value ranges of $e_0$, as illustrated in Table 6. All values of the structural eccentricities are expressed as a percentage of the buildings' plan dimensions $L_i$ ($i = 1, 2$, Figure 14), following the rationale of seismic codes such as EN1998-1 [49].

**Table 6.** Relation assumed between the structural data inserted by the user and the structural eccentricity $e_0$.

|  | Case 1 | Case 2A | Case 2B | Case 3 |
|---|---|---|---|---|
| **Regularity in plan** | Yes | Yes | No | No |
| **Regular distribution of masonries** | Yes | No | Yes | No |
| **Eccentricity** | Low | Medium | | High |
| **$mine_{0i}$/$maxe_{0i}$** | $0.05L_i$/$0.075L_i$ | $0.075L_i$/$0.15L_i$ | | $0.15L_i$/$0.175L_i$ |

As regards the assumptions for the estimation of the ratios $n_{v1}$ and $n_{v2}$, the following three structural data inserted by the user (Figure 7) are considered: (a) The existence (or not) of RC shear walls along the two horizontal directions (axes) x($\equiv$1), y($\equiv$2) if this information is reliably known, (b) the year or the period of the building's construction, and (c) the application (or not) of seismic code provisions for the design of the building. All the structural data described above are considered to define the maximum and the minimum values of $n_{v1}$ and $n_{v2}$. In the current version of the software application, the limit values of $n_{v1}$ and $n_{v2}$ were estimated considering the construction practices followed in Greece. However, an expansion of the software source code in order to cover different limit values for $n_{v1}$ and $n_{v2}$ is relatively easy. The limit values of $n_{v1}$ and $n_{v2}$ inserted in the current version of the software application are presented in Table 7. A different value range is considered according to the period of construction and the seismic code considered for the design (no code, limited requirements, current code provisions). The latter is related to the assumption that relevant code provisions regarding the base shear ratios $n_{v1}$ and $n_{v2}$ are applied.

**Table 7.** Relation between the structural data inserted by the user and the ratios $n_{v1}$ and $n_{v2}$ ($minn_{v1}/maxn_{v2}$).

| Existence of RC Shear Walls | Usage of Seismic Code | Period (Year) of Construction | | | | | |
|---|---|---|---|---|---|---|---|
| | | <1959 | 1959–1984 | 1984–1992 | 1992–2000 | 2000–2010 | >2010 |
| | | - | R/D 1959 [1] | Expansion of R/D 1959 | NEAK [2] | EAK/2000 [3] | EAK/2000 and Eurocodes |
| No | Yes | $minn_v = maxn_v = 0.0$ | | | | | |
| | No | | | | | | |
| Yes | Yes | 0.05/0.15 | 0.1/0.30 | 0.2/0.40 | 0.25/0.45 | 0.275/0.45 | 0.35/0.65 |
| | No | 0.025/0.05 | 0.025/0.05 | 0.05/0.075 | 0.075/0.10 | 0.075/0.10 | 0.10/0.20 |
| Unknown | Yes | 0.0/0.15 | 0.0/0.30 | 0.0/0.40 | 0.0/0.45 | 0.0/0.45 | 0.0/0.65 |
| | No | 0.0/0.05 | 0.0/0.05 | 0.0/0.075 | 0.0/0.10 | 0.0/0.10 | 0.0/0.20 |

[1] R/D: Royal Decree on the Seismic Code for Building Structures [50]. [2] NEAK: The New Greek Antiseismic Regulations [51]. [3] EAK/2000: Greek Seismic Code [52].

Using the ranges of possible minimum and maximum values of structural parameters $n_{v1}$, $n_{v2}$, and $e_0$ (Tables 6 and 7), the eight input vectors for the MFPNN of Equation (3) are automatically formed using the external function "FORM_ANN_INPUT". The combination of all the possible $n_{v1}$, $n_{v2}$, and $e_o$ parameter values (maximum and minimum) results in eight different input vectors used to define eight different classifications of the examined RC building to the three pre-defined SDC (i.e., to the corresponding DS) of Table 3. The worst classification is presented in the panel "RESULTS" (Figure 10).

$$\mathbf{x} = \begin{bmatrix} \mathbf{x}_{seism} \\ \mathbf{x}_{struct} \end{bmatrix} \rightarrow \begin{aligned} [\mathbf{x}_{struct}]_1 &= \begin{bmatrix} H_{tot} \\ mine_0 \\ minn_{v1} \\ minn_{v2} \end{bmatrix}, & [\mathbf{x}_{struct}]_3 &= \begin{bmatrix} H_{tot} \\ mine_0 \\ minn_{v1} \\ maxn_{v2} \end{bmatrix}, & [\mathbf{x}_{struct}]_5 &= \begin{bmatrix} H_{tot} \\ mine_0 \\ maxn_{v1} \\ minn_{v2} \end{bmatrix}, & [\mathbf{x}_{struct}]_7 &= \begin{bmatrix} H_{tot} \\ mine_0 \\ maxn_{v1} \\ maxn_{v2} \end{bmatrix} \\ [\mathbf{x}_{struct}]_2 &= \begin{bmatrix} H_{tot} \\ maxe_0 \\ minn_{v1} \\ minn_{v2} \end{bmatrix}, & [\mathbf{x}_{struct}]_4 &= \begin{bmatrix} H_{tot} \\ maxe_0 \\ minn_{v1} \\ maxn_{v2} \end{bmatrix}, & [\mathbf{x}_{struct}]_6 &= \begin{bmatrix} H_{tot} \\ maxe_0 \\ maxn_{v1} \\ minn_{v2} \end{bmatrix}, & [\mathbf{x}_{struct}]_8 &= \begin{bmatrix} H_{tot} \\ maxe_0 \\ maxn_{v1} \\ maxn_{v2} \end{bmatrix} \end{aligned} \quad (3)$$

Finally, it should be outlined that the parametric analysis described above is not mandatory, since user-defined input values could also be applied. More specifically, if the user can estimate more reliably the minimum and the maximum values of the unknown structural parameters it is possible to insert these values using the GUI and run separately the software application for each one of them. In other words, in this case, the user can perform the required parametric analyses manually.

## 5. Numerical Applications

The current section applies the software developed for both pre-and post-earthquake seismic damage assessment of RC buildings. Two different cases of data availability are considered, namely full data availability for RC buildings used for the pre-earthquake assessment and limited data availability for the case of post-earthquake assessment. The latter is frequently the case in real-time applications, where the need for rapid estimation of the seismic damage level is evident, along with the difficulties for structural data collection. In this case, the parametric investigation considering several scenarios regarding the values of the required structural parameters of the examined RC buildings that are not known is performed (Section 4.4). The ability of the developed software to instantly provide the classification of an RC building to the pre-defined DS, based on the use of the embedded MFPNN, makes the parametric investigation possible, even in real-time analyses.

### 5.1. Software Application for the Pre-Earthquake Assessment of RC Buildings with Full Data Availability

In this case, the software was applied for 30 RC buildings with reliably known structural parameters presented in Table 8 and explained in Figure 14. These buildings were also analyzed through non-linear time history analyses (NTHA), which is the most effective numerical method for estimating the seismic DS (in the present paper using the MIDR seismic damage index). At first, the selected RC buildings were modeled and designed considering the EN1992-1-1 [53] and EN1998-1 [49] recommendations. The buildings, designed with current code provisions, were subsequently assessed by means of NTHA for 65 known seismic excitations extracted from the European Strong-Motion Database [54] and PEER [48]. The ranges of the seismic parameters' values of the 65 selected seismic excitations are given in [33].

**Table 8.** Structural data of the 30 examined known RC buildings used for the prediction of their DS.

|  | Name | $n_{v1}$ | $n_{v2}$ | $H_{tot}$ (m) | $L_x$ (m) | $L_y$ (m) | $e_x$ (m) | $e_y$ (m) |
|---|---|---|---|---|---|---|---|---|
| 1 | SFxy_3 | 0.00 | 0.00 | 9.60 | 13.50 | 10.00 | 0.00 | 0.00 |
| 2 | SFxy_5 | 0.00 | 0.00 | 16.00 | 20.00 | 14.00 | 0.00 | 0.00 |
| 3 | SFxy_7 | 0.00 | 0.00 | 22.40 | 20.00 | 14.00 | 0.00 | 0.00 |
| 4 | SWxy_3 | 0.73 | 0.76 | 9.60 | 15.00 | 10.00 | 0.00 | 0.00 |
| 5 | SWxy_5 | 0.77 | 0.80 | 16.00 | 19.00 | 16.40 | 0.00 | 0.00 |
| 6 | SWxy_7 | 0.57 | 0.64 | 22.40 | 19.00 | 16.40 | 0.00 | 0.00 |
| 7 | SFExy_3 | 0.41 | 0.41 | 9.60 | 15.00 | 15.00 | 0.00 | 0.00 |
| 8 | SFExy_5 | 0.46 | 0.50 | 16.00 | 21.00 | 18.50 | 0.00 | 0.00 |
| 9 | SFExy_7 | 0.43 | 0.46 | 22.40 | 21.00 | 18.50 | 0.00 | 0.00 |
| 10 | SFExFy_3 | 0.43 | 0.00 | 9.60 | 17.00 | 12.50 | 0.00 | 0.00 |
| 11 | SFExFy_5 | 0.41 | 0.00 | 16.00 | 20.00 | 15.00 | 0.00 | 0.00 |
| 12 | SFExFy_7 | 0.38 | 0.00 | 22.40 | 20.00 | 15.00 | 0.00 | 0.00 |
| 13 | SWxFy_3 | 0.77 | 0.00 | 9.60 | 15.00 | 10.00 | 0.00 | 0.00 |
| 14 | SWxFy_5 | 0.68 | 0.00 | 16.00 | 20.00 | 15.00 | 0.00 | 0.00 |
| 15 | SWxFy_7 | 0.51 | 0.00 | 22.40 | 20.00 | 15.00 | 0.00 | 0.00 |
| 16 | AFxy_3 | 0.00 | 0.00 | 9.60 | 13.00 | 9.00 | 0.942 | 0.272 |
| 17 | AFxy_5 | 0.00 | 0.00 | 16.00 | 17.50 | 10.00 | 2.545 | 0.395 |
| 18 | AFxy_7 | 0.00 | 0.00 | 22.40 | 17.50 | 10.00 | 2.35 | 0.420 |
| 19 | AFExy_3 | 0.52 | 0.46 | 9.60 | 13.50 | 9.00 | 4.12 | 2.14 |
| 20 | AFExy_5 | 0.43 | 0.42 | 16.00 | 16.00 | 14.50 | 3.28 | 2.61 |
| 21 | AFExy_7 | 0.37 | 0.36 | 22.40 | 16.00 | 14.50 | 2.98 | 2.35 |
| 22 | AFExFy_3 | 0.47 | 0.00 | 9.60 | 13.50 | 9.00 | 0.71 | 2.11 |
| 23 | AFExFy_5 | 0.38 | 0.00 | 16.00 | 16.00 | 14.50 | 0.45 | 2.61 |
| 24 | AFExFy_7 | 0.35 | 0.00 | 22.40 | 16.00 | 14.50 | 0.45 | 2.45 |
| 25 | AWxFy_3 | 0.64 | 0.00 | 9.60 | 14.50 | 9.00 | 0.30 | 3.51 |
| 26 | AWxFy_5 | 0.69 | 0.00 | 16.00 | 14.00 | 16.00 | 2.80 | 1.11 |
| 27 | AWxFy_7 | 0.65 | 0.00 | 22.40 | 14.00 | 16.00 | 2.76 | 1.20 |
| 28 | AWxy_3 | 0.64 | 0.58 | 9.60 | 13.50 | 10.00 | 5.55 | 3.81 |

**Table 8.** *Cont.*

|  | Name | $n_{v1}$ | $n_{v2}$ | $H_{tot}$ (m) | $L_x$ (m) | $L_y$ (m) | $e_x$ (m) | $e_y$ (m) |
|---|---|---|---|---|---|---|---|---|
| 29 | AWxy_5 | 0.65 | 0.72 | 16.00 | 16.25 | 16.25 | 3.11 | 5.46 |
| 30 | AWxy_7 | 0.59 | 0.67 | 22.40 | 16.25 | 16.25 | 2.79 | 5.27 |

Notation: The names of the buildings code their structural characteristics as follows: x, y = Axes of buildings (referring to directions 1 and 2 respectively); S = <u>S</u>ymmetric structural system along directions 1 (axis x) and 2 (axis y); A = <u>A</u>symmetric structural system; W = <u>W</u>all system according to EN1998-1 (the R/C shear walls receive more than 65% of base shear force); F = <u>F</u>rame system according to EN1998-1 (the R/C shear walls receive less than 35% of base shear force); FE = Dual system equivalent to frame system according to EN1998-1 (the R/C shear walls receive 35–50% of the base shear force); 3, 5, 7 are the number of stories.

All the selected buildings are rectangular in plan and regular in elevation according to the provisions of EN1998-1. It must be noted that for each one of the 30 selected buildings, three different versions were considered regarding their masonry infills: (a) without masonry infills or with masonry infills with a weak contribution to the buildings' seismic response ("Bare Buildings"—"BB"), (b) with strong masonry infills at all stories ("Fully Infilled Buildings"—"FIB") and (c) with strong masonry infills at all stories except the ground story ("Pilotis Buildings"—"PB"). The contribution of the masonry infills to the buildings' seismic response was considered in the framework of NTHA, using the model proposed in [55]. At the design stage, the contribution of the masonry infills was ignored. Thus, the masonry infills were considered only as vertical loads at that stage.

The NTHA performed led to 1950 (=30 × 65) MIDR values. Thus, for the three versions of the 30 selected RC buildings 5850 (=1950 × 3) MIDR values were calculated. Using these values, 5850 classifications to the three pre-defined SDC (i.e., DS) of Table 3 were performed.

The classification of the 5850 generated samples to the three pre-defined SDCs using the results of NTHA constitutes the dataset for the examination of the efficiency of the developed software application. A MATLAB-compatible script was developed to run the application automatically 5850 times using the data of the 90 selected RC buildings and the 65 selected seismic excitations. The classification results based on NTHA and the relevant derived from the software application were subsequently compared. The comparisons are illustrated in terms of confusion matrices (CM) [11,56], which are the most effective tools for the evaluation of the effectiveness of trained MFPNN used for the solution of PR problems.

The CMs for the 5850 samples, corresponding to the 90 known RC buildings (with full data availability, namely reliable knowledge of their structural data), are given in Figure 15.

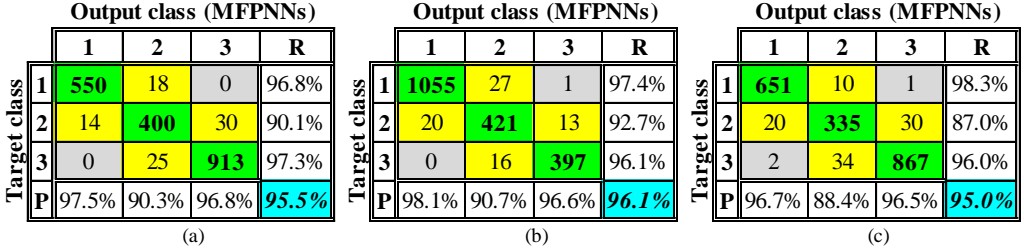

**Figure 15.** CMs of the classifications performed by the MFPPNs embedded in the current version of the software application (**a**) for BB, (**b**) for FIB, and (**c**) for PB.

The performance of the MFPNNs embedded in the current version of the software application is significantly high. However, the most crucial point in the present case is that the implementation of the MFPNNs can be achieved in almost real-time, exploiting the developed GUI capabilities for the input of data for a large number of buildings with reliably known structural parameters.

### 5.2. Software Application for the Post-Earthquake Real-Time Assessment of RC Buildings with Limited Data Availability

In this case, the software was applied to three case study RC buildings with structural parameters that are considered unknown in order to check the reliability of the parametric procedure based on the proposed assumptions described in Section 4.4. In fact, the structural parameters of these buildings are known and presented in Table 9. However, in order to proceed with the reliability check, the structural data of this Table were not directly inserted into the application to investigate the effectiveness of the parametric analyses performed by the external function "FORM_ANN_INPUT" when the structural parameters of the MFPNNs' input vectors are considered unknown (Section 4.4).

**Table 9.** Structural data of the three examined buildings with unknown data used for the prediction of their DS.

| | Name | $n_{v1}$ | $n_{v2}$ | $H_{tot}$ (m) | $L_x$ (m) | $L_y$ (m) | $e_x$ (m) | $e_y$ (m) |
|---|---|---|---|---|---|---|---|---|
| 1 | SFExFy_3 | 0.62 | 0.0 | 9.6 | 10.0 | 15.0 | 0.0 | 0.0 |
| 2 | SFExFy_5 | 0.60 | 0.0 | 16.0 | 10.0 | 15.0 | 0.0 | 0.0 |
| 3 | SFExFy_8 | 0.58 | 0.0 | 25.6 | 10.0 | 15.0 | 0.0 | 0.0 |

The case study buildings were designed according to the provisions of EN1992-1-1 and EN1998-1. The information about the application of these codes for the design of the buildings in Table 9 was considered as known. Three different versions of each one of the three case study buildings were considered, differentiated regarding their masonry infills; i.e., "Bare Buildings"- "BB", "Fully Infilled Buildings"- "FIB", and "Pilotis Buildings"- "PB" (Section 5.1). After the design procedure, the case study buildings were assessed using NTHA for the 65 seismic excitations, also used for the analyses of the 30 known RC buildings presented in the previous section. Based on the analysis results, the calculation of the seismic damage indices (i.e., the MIDR values) and consequently the classifications to the three pre-defined DS of Table 3 were extracted. Since the structural parameters $n_{v1}$, $n_{v2}$, and $e_0$ were considered unknown, the assumptions described in Section 4.4. were followed. To this end, the data required to define the range of the possible values of unknown structural parameters for the three case study buildings were estimated and inserted into the application (Table 10).

**Table 10.** Structural data of the three versions of the three case study buildings.

| Building | Year of Construction | Regularity in Plan | Existence of RC Shear Walls | | Regular Distribution of Masonries | Strong Masonry Infills | Pilotis |
|---|---|---|---|---|---|---|---|
| | | | Dir 1 | Dir 2 | | | |
| SFExFy_3B | >2010 | Yes | Unknown | Unknown | Yes | | |
| SFExFy_5B | >2010 | Yes | Yes | Unknown | No | No | No |
| SFExFy_8B | >2010 | Yes | Unknown | No | No | | |
| SFExFy_3F | >2010 | Yes | Unknown | Unknown | Yes | | |
| SFExFy_5F | >2010 | Yes | Yes | Unknown | No | Yes | No |
| SFExFy_8F | >2010 | Yes | Unknown | No | No | | |
| SFExFy_3P | >2010 | Yes | Unknown | Unknown | Yes | | |
| SFExFy_5P | >2010 | Yes | Yes | Unknown | No | Yes | Yes |
| SFExFy_8P | >2010 | Yes | Unknown | No | No | | |

Notation: The names of the three case study buildings are coded using the symbols used for the 30 known RC buildings in Section 5.1. (Table 8). The characters B, F, P denote the "Bare Buildings", the "Fully Infilled Buildings", and the "Pilotis Buildings".

Considering the structural data of Table 10 and under the assumptions of Section 4.4, the unknown values of the structural parameters $n_{v1}$, $n_{v2}$, and $e_0$ of the three case study buildings fluctuate into the ranges presented in Table 11.

**Table 11.** Assumed ranges of the values of the case study buildings' unknown structural parameters.

| Building | $n_v$ Ratios | | | | Eccentricity $e_0$ | |
| | Dir 1 ($n_{v1}$) | | Dir 2 ($n_{v2}$) | | | |
| | $minn_{v1}$ | $maxn_{v2}$ | $minn_{v1}$ | $maxn_{v2}$ | $mine_0$ | $maxe_0$ |
|---|---|---|---|---|---|---|
| SFExFy_3B | 0.00 | 0.65 | 0.00 | 0.65 | 0.9014 | 1.352 |
| SFExFy_5B | 0.35 | 0.65 | 0.00 | 0.65 | 1.3521 | 2.704 |
| SFExFy_8B | 0.00 | 0.65 | 0.00 | 0.00 | 1.3521 | 2.704 |
| SFExFy_3F | 0.00 | 0.65 | 0.00 | 0.65 | 0.9014 | 1.352 |
| SFExFy_5F | 0.35 | 0.65 | 0.00 | 0.65 | 1.3521 | 2.704 |
| SFExFy_8F | 0.00 | 0.65 | 0.00 | 0.00 | 1.3521 | 2.704 |
| SFExFy_3P | 0.00 | 0.65 | 0.00 | 0.65 | 0.9014 | 1.352 |
| SFExFy_5P | 0.35 | 0.65 | 0.00 | 0.65 | 1.3521 | 2.704 |
| SFExFy_8P | 0.00 | 0.65 | 0.00 | 0.00 | 1.3521 | 2.704 |

The eight input vectors of Equation (3) were formed for each one of the case study buildings and the 65 seismic excitations, using the values of Table 11 as input. Thus, for each one of the three case study buildings, 520 (=65 earthquakes·8 input vector types of Equation (3)) input vectors were created, and 1560 (=520 × 3 versions of buildings) input vectors were introduced to the MFPNNs, embedded in the current version of the developed software application. Consequently, the software was automatically applied 4680 (=1560 × 3 case study buildings) times using an appropriate script. In order to check the reliability of the results, the software was also applied, considering the real values of the structural parameters $n_{v1}$, $n_{v2}$, and $e_0$ (Table 9).

The overall accuracy (OA) index values, i.e., the percentages of the correct classifications of the case study buildings subjected to the 65 selected seismic excitations into the three pre-defined SDC (DS), as calculated by the MFPNNs, are presented in Figure 16. These OA index values correspond to each one of the eight input vectors $\mathbf{Xi}$ (i = 1–8) of Equation (3) [OA($\mathbf{Xi}$)] as well as to the input vectors $\mathbf{XR}$, which include the actual values of the structural parameters $n_{v1}$, $n_{v2}$, and $e_0$ ([OA($\mathbf{XR}$)] – black bars).

The main conclusion that arises from the study of Figure 16 is that the parametric procedure, based on the consideration of eight input vectors $\mathbf{Xi}$ (Equation (3)), can effectively approach the percentages of the correct classifications extracted using the input vectors $\mathbf{XR}$. More specifically, in the case of BB, the differences between the OA($\mathbf{Xi}$) index values and the corresponding OA($\mathbf{XR}$) index values (i.e., the Δ(OA) values in Figure 16) fluctuate between −1.8% and −18.3%, while the corresponding lower and upper values of differences in the cases of FIB and PB are −0.6%/−9.4% and −1.9%/−19.9%, respectively. Bearing in mind that the feasible level of accuracy of the RVI methods is generally low, the abovementioned deviations between the OA($\mathbf{Xi}$) and the OA($\mathbf{XR}$) index values (of the order of 10–15%) can be considered acceptable (OA > 70% in any case). However, it must be outlined that the considered value ranges of the unknown structural parameters $n_{v1}$, $n_{v2}$, and $e_0$ strongly affect the deviation between the OA($\mathbf{Xi}$) and the OA($\mathbf{XR}$) index values. In the present paper, the ranges summarized in Table 11 were used. The software proposed enables the selection of different ranges for the values on the unknown structural parameters based on the user's expertise. Further investigation of these ranges is possible but out of the scope of the present paper.

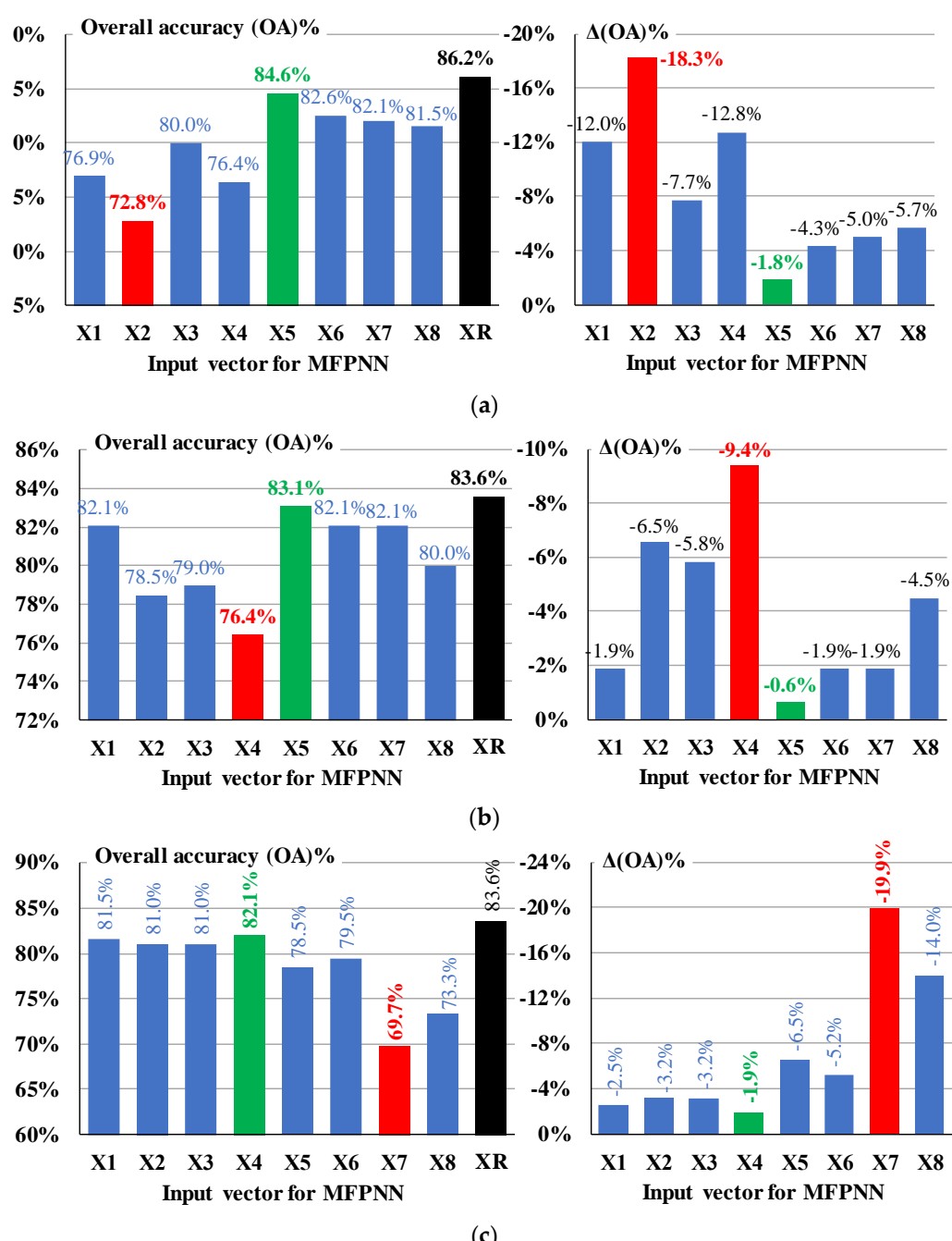

**Figure 16.** OA index values of the classifications extracted by the used MFPNNs considering input vectors with assumed values for the unknown structural parameters: (**a**) for BB, (**b**) for FIB, and (**c**) PB.

Finally, the CMs corresponding to the classifications performed using the input vectors **XR** and the input vectors **Xi** that provide the min[Δ(OA)] and the max[Δ(OA)] of Figure 16 are presented in Figure 17. The study of the CMs extracted using these **Xi** led to the conclusion that the vast majority of the recall (R-index) and the precision (P-index) indices are close to the corresponding values of the CMs extracted using **XR** as input vectors. Thus, the classifications of the three case study buildings to the three pre-defined DS extracted using the input vectors **Xi** are not significantly different from the corresponding classifications extracted using the input vectors **XR** with the actual values of the structural parameters $n_{v1}$, $n_{v2}$, and $e_0$. However, as mentioned above, the approach based on the use of input vectors **Xi** (if the input vectors **XR** are unknown) can be significantly improved

if the users' expertise can lead to a more reliable estimation of the unknown structural parameters $n_{v1}$, $n_{v2}$, and $e_0$ than the one presented in Table 11. To this end, the software is developed open to including user-defined values.

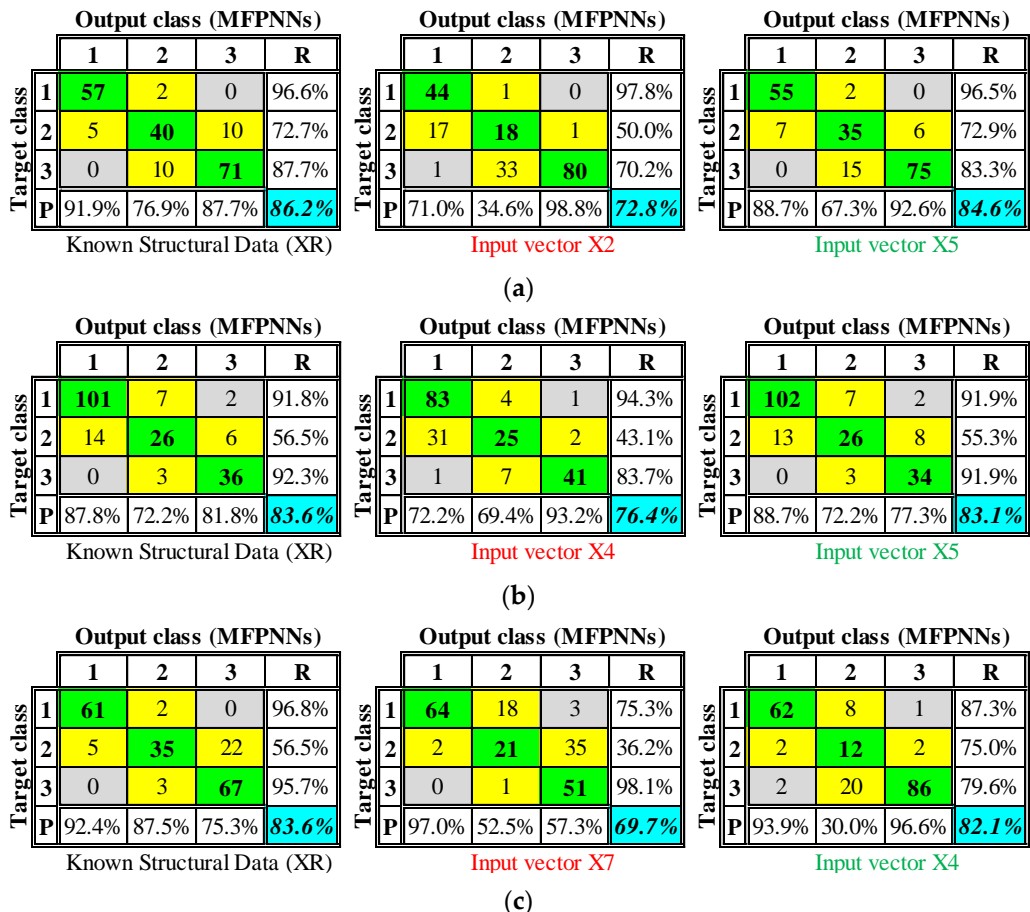

**Figure 17.** CMs correspond to the classifications extracted by the used MFPNNs considering input vectors with assumed values for the unknown structural parameters: (**a**) for BB, (**b**) for FIB, and (**c**) PB.

## 6. Conclusions

The scope of the current paper is to emerge and highlight the functions, the applicability, and the advantages of a new software application developed for the rapid seismic damage assessment of RC buildings. The developed software is intended to instantly provide assessment results and is used for both pre-and post-earthquake assessments. Whereas the required seismic data can be instantly available either from strong-motion databases or in real-time after a seismic event, the level of availability of the structural data (parameters) is low due to the limited time and limited resources.

The selected ANN-based method for the seismic damage assessment of RC buildings is a method based on the definition and the solution of pattern recognition (PR) problems. More specifically, by formulating the structures' seismic damage assessment problem (compatible with "first stage simplified methodologies") in terms of PR problems and of ANNs' function, the software application extracts predictions regarding the seismic damage state (DS) of RC buildings in real-time. This DS is quantified considering three seismic damage classes (SDC) defined using specific threshold values of a seismic damage index (SDI). In the current paper, the maximum interstory drift ratio (MIDR), which is extensively used in research studies, was selected as SDI. The selection of three DSs was in line with the DS proposed in RVIs, in order to be directly applicable. The seismic and structural parameters were selected as input parameters for the ANNs in the current version of the

proposed application; however, they can be easily changed in future versions due to the flexibility of the source code developed.

The current version of the developed software application was evaluated by predicting the DS of two general categories of RC buildings, namely buildings with reliable known values of the selected structural parameters and buildings with unknown (or no-reliable known) values of the selected structural parameters. In both cases, the effectiveness of the proposed application regarding its capability to extract reliable results in real-time was highlighted. However, it must be noted that the accuracy of the results extracted by the application depends on the effectiveness of the ANNs used. The latter is not the subject of the present paper since already trained ANNs were used. In line with the above, the usage of user-defined ANNs trained with different databases is also possible.

In any case, the capability for direct (in real-time) and reliable estimation of the DS of a building or building inventories in the pre-earthquake phase or after a strong earthquake, avoiding time-consuming modeling and analyses, without exclusively requiring detailed RVI input data (implementing parametric analyses), renders the software developed a valuable calculational tool available to the authorities either for retrofit prioritization in the pre-earthquake phase or emergency planning after a strong earthquake event.

**Author Contributions:** Conceptualization, K.M.; methodology, K.M. and S.S.; software, K.M. and O.M.; validation, S.S. and O.M.; investigation, K.M., S.S. and O.M.; writing—original draft preparation, K.M.; writing—review and editing, S.S. and O.M.; project administration, K.M.; funding acquisition, S.S. and O.M. All authors have read and agreed to the published version of the manuscript.

**Funding:** This research was funded by the European Regional Development Fund of the European Union and Greek national funds through the Operational Program Competitiveness, Entrepreneurship and Innovation, under the call RESEARCH CREATE INNOVATE—Second Round (project code: T2EDK-03412).

**Institutional Review Board Statement:** Not applicable.

**Informed Consent Statement:** Not applicable.

**Data Availability Statement:** Data are only available after contacting the authors.

**Conflicts of Interest:** The authors declare no conflict of interest.

## Abbreviations

The following abbreviations are used in this manuscript:

| | |
|---|---|
| ANN | Artificial Neural Network |
| BB | Bare Building |
| BIM | Building Information Modelling |
| CM | Confusion Matrix |
| DS | Damage State |
| EDP | Engineering Demand Parameters |
| EPPO | Earthquake Planning and Protection Organization (Greece) |
| FEMA | Federal Emergency Management Agency |
| FIB | Fully Infilled Building |
| FSRVS | First Stage Rapid Visual Screening |
| GUI | Graphic User Interface |
| MB | Masonry Building |
| MFPNN | Multilayered Feedforward Perceptron Neural Networks |
| MIDR | Maximum Interstory Drift Ratio |
| ML | Machine Learning |
| NTHA | Non-linear Time History Analysis |
| OA | Overall Accuracy |
| P | Precision (P-index) |
| PB | Buildings with Pilotis |
| PEER | Pacific Earthquake Engineering Research |

| | |
|---|---|
| PR | Pattern Recognition |
| R | Recall (R-index) |
| RVI | Rapid Visual Inspection |
| RVS | Rapid Visual Screening |
| SDI | Seismic Damage Index |
| SDC | Seismic Damage Class |

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
