# Peer review of "A Rapid Seismic Damage Assessment (RASDA) Tool for RC Buildings Based on an Artificial Intelligence Algorithm"

_applsci, doi:10.3390/app13085100_

Round 1

Reviewer 1 Report

The paper presents a software to perform a rapid damage assessment for RC buildings, based on artificial neural network. The paper is interesting and should be accepted for publication. Nevertheless, some aspects should be clarified before to re-evaluate the paper for publication. 

- The introduction is good, but authors should enlarge the part related to the fast assessment of buildings. In general, several methodologies are available, from the mechanical to empirical ones, up to the RVS methods. I suggest to check an existing work and check the references therein.

- Section 2 is to little. I suggest to increase it by adding some existing works on RVS methods, being this the methodology employed herein

- Section 3 also should be characterized by an overall visions of the methodologies to use ML in this field. Few importance was given to the data, which are the base of each training/test/validation method. As for example, in 10.3390/data7010004, specific information is provided on the data used for training an analogous software as the one herein proposed

- The code description is very well written, as well as its way to be applied. The real problem I see is that it was not presented a real case study application (it seems that the application is numerical). This could give value to your proposal. 

-In the en, please, check the english, which in some parts could be more fluent

Reviewer 2 Report

The study titled “Α Rapid Seismic Damage Assessment (RASDA) tool for RC Buildings based on an Artificial Intelligence Algorithms” is a software development and application tool based on ANN to assess RC buildings subjected to earthquake excitation. For this, 30 RC building were examined with known Structural data and three RC building with unknown data used for the prediction of damage states. Although paper seems interesting, it still needs considerable revisions which are provided for authors as follows:

1.      Introduction is superficial. There are various tools/methods developed for rapid assessment RC or various building types. For this reason, a first paragraph should contain general literature studies including not only RC but also various building types to emphasize the importance of the topic. Then, aim of this study and what current study fills the gap between the rivalry methods should be introduced.

Some recent studies using different methodologies for various buildings are recommended to be quoted in introduction (more recent references must be included):

Fuzzy rule based seismic risk assessment of one-story precast industrial buildings. Earthquake Engineering and Engineering Vibration, https://doi.org/10.1007/s11803-019-0526-5.

Fast Seismic Assessment of Built Urban Areas with the Accuracy of Mechanical Methods Using a Feedforward Neural Network, Sustainability, https://doi.org/10.3390/su14095274

Machine learning network suitable for accurate rapid seismic risk estimation of masonry building stocks, https://doi.org/10.1007/s11069-022-05553-y

2.      In the study, there are many notations. For this reason, it would be useful to add nomenclature list for clear understanding of readers.

3.      In table 2, the list of seismic parameters for MFPNNs was provided. Some parameters have very different characteristics of earthquakes while some of them have no clear correlation with seismic performance. Why did authors preferred to use such a divergent seismic parameters?

4.      Authors claimed that they developed “Α Rapid Seismic Damage Assessment“ tool, but they did not include “Collapse/Destruction” SDC not RVI method. It can be understood that it is hard to classify no damage but exception of “Collapse” seems unreasonable. Authors should provide concrete explanations about this issue.

5.      Authors claimed that they have provided an executable file. However, reviewer did not see any file for observation. File should be provided with the manuscript or a proper link to download file is required.

6.      Is there any format (2 colums, 4 coloumns file etc.) for the input acceleration data. Needs clarification.

7.      In line, 279, What is the “E.P.P.O.”, abbreviations check.

8.      “Spectrums” should be replaced by “Spectra”.

9.      In some places, authors notated in figures nvx and nvy, but nv1 and nv2 in the text. Although they mean the same, a unique language is more appropriate for the integrity of manuscript.

10.   In table 8, the values for nvx and nvy should be wrong! I think authors forgotten to convert values from percentage to numeric ones.

11.   In section 4.2.4, the option b, line 335-339 is nor clear. More specific information should be provided for the potential users.

12.   It can be understood from the section 5 that developed methodology is actually based on 30 theoretically designed and analyzed RC buildings. Buildings were exposed to 65 earthquakes using different databases which the characteristics of earthquakes are not known. How were they selected? The selection criteria of records for the NLTH are not known? This section is important since whole method based on this, so how did authors decide that selected earthquake was reliable to make damage assessment of selected buildings?

13.   The determination of nvx, nvy is not clear. For the rapid assessment tools calculation of deterministic values especially base shear is quite hard and it is not practical for such tools. Method offers to use unknow parameters but base shear is very effective parameter on MIDR and assumptions with unknown values might decreases the reliability of the tool and makes the assessment tool  hypothetical and avoids its applicability widespread. Although authors advised to use multiple input vectors to account this variability, it can produce very irrelevant results. Needs more explanations about this issue

14.   Although adapted methods are flexible, developed method based on adapted buildings and buildings were design according to Greece seismic codes which implies the results in local scale. This situation lacks the novelty of study. For this reason, one or two paragraphs including pros and cons of developed method should be discussed in detail. Possible future improvement should be also clarified.

15.   In figure 16, effect of input vectors on the accuracy of developed method is introduced. It seems from the figure that different input vectors have different and important effect on the quality of accuracy. Did authors investigate the reasons behind it? Needs explanation.

Round 2

Reviewer 2 Report

Satisfactory comments are made to Reviewers' questions.